# Toxicity of Metal Oxide Nanoparticles: Looking through the Lens of Toxicogenomics

**DOI:** 10.3390/ijms25010529

**Published:** 2023-12-30

**Authors:** Andrey Boyadzhiev, Dongmei Wu, Mary-Luyza Avramescu, Andrew Williams, Pat Rasmussen, Sabina Halappanavar

**Affiliations:** 1Environmental Health Science and Research Bureau, Health Canada, Ottawa, ON K1A 0K9, Canada; andrey.boyadzhiev@hc-sc.gc.ca (A.B.); dongmei.wu@hc-sc.gc.ca (D.W.); mary-luyza.avramescu@hc-sc.gc.ca (M.-L.A.); andrew.williams@hc-sc.gc.ca (A.W.); pat.rasmussen@hc-sc.gc.ca (P.R.); 2Department of Biology, University of Ottawa, Ottawa, ON K1N 6N5, Canada; 3Department of Earth and Environmental Sciences, University of Ottawa, Ottawa, ON K1N 6N5, Canada

**Keywords:** nanotoxicology, nanomaterials, canonical pathways, omics, enrichment analysis, potency ranking, BMC modelling

## Abstract

The impact of solubility on the toxicity of metal oxide nanoparticles (MONPs) requires further exploration to ascertain the impact of the dissolved and particulate species on response. In this study, FE1 mouse lung epithelial cells were exposed for 2–48 h to 4 MONPs of varying solubility: zinc oxide, nickel oxide, aluminum oxide, and titanium dioxide, in addition to microparticle analogues and metal chloride equivalents. Previously published data from FE1 cells exposed for 2–48 h to copper oxide and copper chloride were examined in the context of exposures in the present study. Viability was assessed using Trypan Blue staining and transcriptomic responses via microarray analysis. Results indicate material solubility is not the sole property governing MONP toxicity. Transcriptional signaling through the ‘HIF-1α Signaling’ pathway describes the response to hypoxia, which also includes genes associated with processes such as oxidative stress and unfolded protein responses and represents a conserved response across all MONPs tested. The number of differentially expressed genes (DEGs) in this pathway correlated with apical toxicity, and a panel of the top ten ranked DEGs was constructed (Hmox1, Hspa1a, Hspa1b, Mmp10, Adm, Serpine1, Slc2a1, Egln1, Rasd1, Hk2), highlighting mechanistic differences among tested MONPs. The HIF-1α pathway is proposed as a biomarker of MONP exposure and toxicity that can help prioritize MONPs for further evaluation and guide specific testing strategies.

## 1. Introduction

Engineered metal oxide nanoparticles (MONPs) are increasingly investigated for their applications in engineering and biomedical fields due to their attractive physical and chemical properties. MONPs have been used in the production of electronics, energy generation and catalysis, biomedicine, and have even seen incorporation into personal care products and foods [1,2,3]. Market research indicated that the global MONP market was valued at 0.9 billion USD in 2021 and is expected to double in value by 2030 [4]. This has resulted in an increased risk of human exposure to these materials in occupational settings, including through the pulmonary system when handling the dry material or via the processing of MONP-containing products [5,6]. However, the human health impacts of exposure to these materials are not entirely understood.

Epidemiological or cohort studies focusing on human pulmonary exposure to engineered MONPs to date are few, but a handful of studies have shown that workers involved in the manufacturing of titanium dioxide (TiO_2_) and human volunteers exposed to engineered zinc oxide (ZnO) nanoparticles (NPs) present elevated levels of markers of cardiopulmonary and systemic toxicity, respectively [6]. For workers employed in TiO_2_ NP manufacturing facilities, elevated levels of leukotrienes (markers of inflammation) in exhaled breath condensate were shown to be positively associated with exposure to TiO_2_ NPs [7,8]. In another study, in a Chinese TiO_2_ manufacturing plant, where 39% of the airborne particle fraction was nanoscale, increased levels of surfactant protein D (SP-D, a marker of inflammation), superoxide dismutase and malondialdehyde (indicative of oxidative stress), and pro-inflammatory markers interleukin-6 and interleukin-1β were observed in the serum of exposed workers [9]. With respect to ZnO exposure, human volunteers exposed to 0.5–2 mg/m^3^ ZnO NPs (primary particle size of 10 nm) via inhalation resulted in symptoms of systemic toxicity known as metal fume fever, including elevated body temperature and elevated levels of acute phase proteins and neutrophils in the serum [10]. Systemic symptoms of toxicity became clear at and above 1 mg/m^3^ ZnO NPs, which is below international workplace exposure limits for respirable ZnO dust (available through the GESTIS—Substance Database, https://limitvalue.ifa.dguv.de/, accessed on 1 October 2023).

Activities such as welding can lead to the generation of incidental MONPs. Welding fumes have been classified as Group 1 carcinogens by the International Agency for Research on Cancer, based on evidence presented in over 30 occupational and population cohort studies showing an increased risk of lung cancer in welders and workers exposed to welding fumes [11]. The composition of welding fumes is heterogeneous, but research has shown that MONPs are present within lesions found in the lungs of welders and are at least partially responsible for the lung pathologies reported in these individuals [12]. While limited in scope, the above studies show that there is a human health risk associated with pulmonary exposure to MONPs and that the potential toxicity is not the same across MONP types.

In both animal and cell model systems, exposure to MONPs is shown to result in varying degrees of toxicity (recently reviewed in [1,13,14,15,16,17,18,19]). For example, C57BL/6 mice exposed to low and moderate doses of insoluble TiO_2_ NPs via single intratracheal instillation developed an acute inflammatory response 24 h post-exposure, which was resolved by the 28 day post-exposure timepoint [20], whereas single intratracheal instillation of high doses of TiO_2_ NPs in C57BL/6 mice resulted in a subtle fibrotic response 90 days post-exposure [21]. Acute inflammation is also a hallmark of pulmonary exposure to soluble MONPs such as ZnO and copper oxide (CuO), which have been shown to proceed to fibrotic or emphysemic outcomes depending on the model organism, method of exposure employed, and duration of exposure. In rats exposed via nose-only inhalation to CuO NPs, acute lung inflammation, emphysema, and gene expression changes related to inflammation were noted one day post-exposure, which mostly resolved after a 22 day recovery period [22,23]. On the contrary, intranasal administration of CuO NPs in C57BL/6 mice resulted in a fibrotic outcome 21 days post-exposure [24], which was also observed following intratracheal instillation of ZnO NPs in C57BL/6 mice 2 months post-exposure [25].

In cell model systems, cytotoxicity and oxidative stress have been commonly reported after exposure to MONPs, with soluble MONPs generally presenting stronger responses compared to insoluble or poorly soluble MONPs [26,27,28]. In vitro mechanistic research has shown that MONPs exert their toxicity through (1) the production of reactive oxygen species (ROS), (2) particle dissolution and subsequent metal ion toxicity, and (3) direct interaction between the MONP and cellular components such as membranes or organelles [16,19,29,30]. These mechanisms can then lead to DNA damage, homeostasis disruption, organelle damage, pro-inflammatory signaling, and calcium signaling alterations. The specific mechanisms and severity of adverse outcomes are influenced by the physicochemical properties of the MONPs, one of the most important of which is particle solubility in biological environments.

Previous research has shown that MONP dissolution is a function of time, particle concentration, particle size, chemical composition, and the solvent medium [31,32]. Due to the time-dependent nature of dissolution, the toxicity of soluble MONPs is a function of both the particulate and dissolved species, whereas MONPs with negligible solubility induce toxicity through interactions with the particle surface. A knowledge gap exists, however, concerning the contribution of the dissolved and particulate species to response over timepoints relevant to MONP dissolution. It is also not completely understood if soluble, poorly soluble, and insoluble MONPs act via similar toxicity mechanisms. Toxicogenomics (application of genomics technologies to study the adverse effects of exposure to substances) tools are particularly well suited for filling this data gap, as they provide mechanistic toxicity information through the analysis of the proteome, genome, or transcriptome of a model system. Recently, using in vitro global transcriptional analysis and cytotoxicity monitoring, Boyadzhiev et al., 2021 demonstrated that, for CuO NPs showing moderate solubility in cell culture medium, the toxicity is potentiated by the particle size, even though apical and molecular responses may be qualitatively similar between NP and dissolved Cu exposures [33]. Furthermore, the same authors recently examined the DNA damage potential of ZnO, CuO, and TiO_2_ NPs, associated microparticles (MPs), and metal chloride salts and showed that ZnO NPs and MPs behaved similarly to dissolved Zn with respect to DNA damage induction, whereas significant differences were observed between CuO forms, with only NPs inducing DNA damage compared to MPs or dissolved Cu [34]. For the TiO_2_ NPs of negligible solubility tested, silica and silica-alumina coatings and a rutile crystal structure potentiated a muted DNA damage response. These studies showed that (1) soluble MONPs can induce pronounced cytotoxic, transcriptional, and genotoxic responses, but the relative influence of the dissolved and particulate species is chemical compound dependent, and (2) the surface chemistry of MONPs exhibiting negligible solubility is important to their response. Additional research is necessary to ascertain whether these observations hold true for MONPs in a broader sense, in which case prioritization and grouping strategies for nanomaterial risk assessment may be expedited.

The objective of the present study was to assess the mechanisms of toxicity of MONPs in lung cells and the influence of material solubility on response. For this purpose, mouse lung epithelial cells (FE1) were exposed to ZnO, nickel oxide (NiO), aluminum oxide (Al_2_O_3_), and TiO_2_ MONPs with varying solubility in cell culture medium, as well as metal chloride and bulk (metal oxide microparticle; MOMP) analogues for 2–48 h. Cytotoxicity and global transcriptional changes were assessed via trypan blue staining and microarray analysis, respectively. Benchmark concentration (BMC) modeling was used to delineate differences in potency between the MONPs and their different forms. Previously published viability and transcriptomic responses of CuO NPs, MPs, and CuCl_2_ from Boyadzhiev et al., 2021, were used for comparison with the results of the present study.

## 2. Results

### 2.1. Exposure Characterization

For CuO, ZnO, and TiO_2_ NPs, transmission electron microscopic (TEM) imaging was previously conducted [33,34], and the results showed their measured particle sizes to be <50 nm. The TEM sizing of both NiO and Al_2_O_3_ NPs also showed mean measured particle sizes <50 nm (Table 1, Appendix A).

The aspect ratio of the measured particles showed that four out of five MONPs had a spherical morphology, while Al_2_O_3_ NPs exhibited a rod-like morphology (Table 1). Dynamic light scattering (DLS) and electrophoretic light scattering (ELS) analyses showed that mean aggregate sizes did not exceed 450 nm, and all particle suspensions presented a negative surface charge in cell culture medium based on zeta potential (Table 1). With respect to the distribution of aggregate sizes, the polydispersity index (PDI) ranged from 0.23 (TiO_2_ NPs) to 0.52 (ZnO NPs), with the CuO NPs previously examined producing the most monodisperse suspension with a PDI of 0.174 (Table 1).

With respect to MOMPs, scanning electron microscopy (SEM) images showed a roughly spherical morphology for NiO and Al_2_O_3_ MPs (Appendix A), which was also reported for CuO and TiO_2_ MPs, while ZnO MPs had a fraction of rod-shaped particles present [33,34]. Particles under 100 nm in size could be seen within aggregates of NiO MPs (Appendix A), as well as for ZnO and TiO_2_ MPs [34].

For the metal chloride solutions, both ZnCl_2_ and NiCl_2_ were soluble in cell culture medium, but not AlCl_3_, which was soluble in water but immediately formed a precipitate in cell culture medium (Appendix A).

Dissolution testing for all relevant MONPs and MOMPs was carried out by Avramescu et al., 2020 [31] and Avramescu et al., 2023 [32], the results of which were used to inform the toxicity testing conducted in the present study (Appendix A). MONPs were grouped as negligible (<1% dissolved), low (1–10% dissolved), moderate (10–70% dissolved), or highly soluble (>70% dissolved) according to Avramescu et al., 2023 [32], which was based on criteria reported in ENV/JM/MONO (2015) [35]. In cell culture medium, at a low initial concentration of 10 µg/mL, % dissolution varied in the following order: ZnO NPs (94.5%, high), CuO NPs (12.6%, moderate), Al_2_O_3_ NPs (1.11%, low), and negligible for NiO NPs and TiO_2_ NPs (0.94% and 0.17%, respectively). At a high initial concentration of 100 µg/mL, % dissolution was moderate for ZnO NPs (19.3%) and CuO NPs (51.6%), low for NiO NPs (1.81%), and negligible for Al_2_O_3_ NPs and TiO_2_ NPs (0.73% and 0.045%, respectively). For bulk MOMP analogues at 100 µg/mL, dissolution varied from moderate for ZnO MPs (11.8%), to low for CuO MPs (1.51%), and negligible for NiO, Al_2_O_3_, and TiO_2_ NPs (<0.07%).

### 2.2. Trypan Blue Viability Analysis

Cytotoxicity following exposure to the four MONPs, MOMPs, and three metal chlorides was assessed using the trypan blue exclusion method (Appendix A). In general, the observed response was concentration-dependent for all compounds that induced a reduction in cell viability.

At the 2 h post-exposure timepoint, NiO NPs significantly reduced percent viability down to 74, 74, and 70% at 10, 25, and 50 µg/mL concentrations. A small but significant reduction in viability, down to 90 and 85% at 25 and 50 µg/mL, respectively, was seen in cells exposed to Al_2_O_3_ NPs. Similarly, for TiO_2_ NPs, a significant reduction in viability down to 88% was observed after exposure to 100 µg/mL. No other exposures at the 2 h timepoint resulted in a significant percent viability reduction.

At the 24 h timepoint, 5 µg/mL of ZnO NPs, MPs, and 8.5 µg/mL ZnCl_2_ resulted in a significant reduction in percent viability down to 77, 72, and 90%, respectively. A significant reduction in percent viability (74%) was observed in cells exposed to 80 µg/mL NiCl_2_. NiO NPs presented a non-significant decrease in viability, from 78–74% after exposure to 5–50 µg/mL concentrations, compared to matched controls, which also showed cytotoxicity (85% viability). With respect to Al materials, 25 and 50 µg/mL Al_2_O_3_ NPs significantly reduced viability down to 87 and 82%, respectively, while Al_2_O_3_ MPs and AlCl_3_ did not affect cell viability. Lastly, for TiO_2_ NPs, small but significant reductions in percent viability were seen after exposure to 25 and 100 µg/mL (89%, 85%) but not after exposure to 50 µg/mL (91%). Cells exposed to TiO_2_ MPs did not show any change in viability at this timepoint.

At 48 h post-exposure, 5 µg/mL of ZnO NPs, MPs, and 8.5 µg/mL ZnCl_2_ significantly reduced viability down to 71%, 52%, and 82%, respectively. Both NiCl_2_ and NiO NPs induced significant reductions in percent viability down to 67 and 45% after exposure to 40 and 80 µg/mL NiCl_2_, and 74 and 52% after exposure to 25 and 50 µg/mL NiO NPs, with no response for NiO MP-exposed cells. Response to Al forms was the same as the 24 h timepoint, with 25–50 µg/mL NPs resulting in a significant reduction in percent viability down to 88–86%, and with no response seen with respect to Al_2_O_3_ MPs and AlCl_3_. Finally, exposure to 50–100 µg/mL TiO_2_ NPs resulted in a significant reduction in viability to 85–83%, with no response seen for TiO_2_ MPs.

In addition to percent viability, the differences in the density of viable cells for exposed and time-matched medium controls were computed (Figure 1). For comparative purposes, the viability data pertaining to FE1 cells exposed for 2–48 h to CuO NPs, MPs, and CuCl_2_ reported in Boyadzhiev et al., 2021 [33] was analyzed alongside the four metal oxides investigated in this study.

At the 2 h timepoint, no significant differences in the density of viable cells were seen, although NiO NP-exposed cells showed decreases in percent viable cell density vs. medium control with 84, 68, 57, and 64% at 5–50 µg/mL concentrations.

At the 24 h timepoint, 5 µg/mL of ZnO NPs, MPs, and 8.5 µg/mL of ZnCl_2_ resulted in significant reductions in viable cell density down to 48%, 16%, and 22% of the time-matched medium control, respectively. For Cu materials previously investigated, cells exposed to 10 and 25 µg/mL NPs showed a significant reduction in the density of viable cells down to 62% and 10% of the medium control, respectively, whereas cells exposed to 54 µg/mL CuCl_2_ significantly reduced viable cell density to 74%, with no response seen for CuO MPs. For cells exposed to Ni materials, a 50 µg/mL concentration of NiO NPs and MPs resulted in significant reductions in viable cell density (60% and 67% of medium control for NPs and MPs, respectively), while 40 and 80 µg/mL NiCl_2_ resulted in significant reductions down to 76% and 42%, respectively. Exposure to Al materials at the 24 h timepoint resulted in reductions in viable cell density down to 83 and 77% of time-matched medium control for 25 and 50 µg/mL NPs and 77% for 50 µg/mL MPs, with no difference seen with respect to AlCl_3_. Finally, for TiO_2_ particles, exposure to 50 and 100 µg/mL of NPs significantly reduced the viable cell density down to 79% and 67% of the medium control in a concentration-dependent manner, while MPs did not have an impact.

At the 48 h timepoint, 5 µg/mL ZnO NPs, MPs, and 8.5 µg/mL ZnCl_2_ resulted in significant reductions in the density of viable cells (40%, 2%, and 6% of time-matched medium controls, respectively). For Cu materials previously investigated, significant reductions in viable cell density were noted down to 59, 7, and 1% of the medium control for 5–25 µg/mL concentrations, while the 54 µg/mL concentration of CuCl_2_ significantly reduced viable cell density to 31% of the medium control, with CuO MPs not affecting response at any tested concentration. For cells exposed to Ni materials, 10–50 µg/mL of NiO NPs significantly reduced the viable cell density to 63, 44, and 24% of the medium control, as did 50 µg/mL of NiO MPs (72%), while NiCl_2_ significantly reduced the viable cell density to 45 and 18% of the medium control at 40–80 µg/mL. With respect to Al materials, 50 µg/mL of Al_2_O_3_ NPs resulted in a significant reduction in viable cell density down to 79% of the medium control, while 10, 25, and 50 µg/mL of Al_2_O_3_ MPs significantly reduced the viable cell density to 85, 87, and 86% of the medium control, which was not concentration-dependent, and 47–118 µg/mL of AlCl_3_ featured a mild but significant reduction in viable cell density down to 79% for both concentrations. Lastly, for cells exposed to 10–100 µg/mL TiO_2_ NPs (but not MPs), significant and concentration-dependent reductions in viable cell density to 87, 73, 69, and 57% of the time-matched medium control were noted.

### 2.3. Differentially Expressed Genes

All tested metal oxide and metal chloride samples presented concentration and time-dependent transcriptional responses, as assessed by the number of DEGs at each timepoint (Figure 2).

With respect to Zn compounds, response to all three forms began at the earliest timepoint of 2 h. There were 1, 4, and 608 DEGs at 0.5, 1, and 5 µg/mL of ZnO NPs, 46 DEGs at 5 µg/mL of ZnO MPs, and 4, 49, and 346 DEGs at 1.7, 5.1, and 8.5 µg/mL of ZnCl_2_. At 24 h, the response increased for all three Zn forms with 1, 5, and 1942 DEGs, 0, 1, and 8367 DEGs, and 3, 184, and 6562 DEGs for ZnO NPs, MPs, and ZnCl_2_, respectively. At 48 h post-exposure, the 0.5 and 1 µg/mL concentrations of ZnO NPs and MPs showed negligible responses, with less than 10 DEGs in each group, while the high concentration of 5 µg/mL induced 2130 and 7907 DEGs for NPs and MPs, respectively. With respect to ZnCl_2_ at the 48 h timepoint, a concentration-dependent response was seen with 3, 371, and 6634 DEGs at 1.7, 5.1, and 8.5 µg/mL concentrations.

A strong transcriptional response was observed for Ni compounds starting at 24 h, with NiCl_2_ inducing more DEGs than NPs or MPs. At the early timepoint of 2 h, both NiO NPs and MPs induced less than 10 DEGs, while the 80 µg/mL concentration of NiCl_2_ induced 14 DEGs. At 24 h, a concentration-dependent response was observed for NPs, MPs, and NiCl_2_, with 0, 25, 554, and 2048 DEGs noted for 5–50 µg/mL NPs, 15, 119, and 335 DEGs observed for 10–50 µg/mL MPs, and 0, 1699, and 2640 DEGs noted for 1.6–80 µg/mL NiCl_2_. Similarly, an increase in DEGs was observed for NPs, MPs, and NiCl_2_ at 48 h, with 2, 63, 1084, and 3668 DEGs for 5–50 µg/mL NPs, 4, 122, and 422 DEGs for 10–50 µg/mL MPs, and 0, 2640, and 4448 DEGs for 1.6–80 µg/mL NiCl_2_.

The Al compounds tested induced the lowest transcriptomic responses out of the four metal oxide materials investigated in this study. At 2 h, all forms and concentrations of Al induced a negligible transcriptional response with less than 10 DEGs. At 24 h, a concentration-dependent response was seen for all three Al compounds, with 0, 2, and 72 DEGs at 10–50 µg/mL NPs, 8, 34, 169 DEGs at 10–50 µg/mL MPs, and 1, 21, 87 DEGs for 2.37–118 µg/mL AlCl_3_ exposed cells. At 48 h there was an increase in response, with 13, 72, 211 DEGs seen for 10–50 µg/mL NPs, 6, 18, 154 DEGs noted for 10–50 µg/mL MPs and 0, 189, 537 DEGs noted for 2.37–118 µg/mL AlCl_3_.

Finally, with respect to TiO_2_ particles, NPs induced a more pronounced DEG response than MPs at all timepoints assessed. At 2 h, a subtle concentration-dependent response with 6, 8, and 19 DEGs was observed for 25, 50, and 100 µg/mL TiO_2_ NPs, but not after exposure to TiO_2_ MPs, which induced 11, 4, and 4 DEGs. At 24 h, the response to both NPs and MPs increased, with 298, 702, and 2299 DEGs for NPs and 4, 37, and 153 DEGs for MPs at 25, 50, and 100 µg/mL concentrations. At 48 h, the response continued to increase, with 606, 1125, and 2127 DEGs and 14, 129, and 358 DEGs noted for 25–100 µg/mL concentrations of NPs and MPs, respectively.

### 2.4. Enriched Canonical Pathways

The total number of enriched pathways closely followed the trends observed for DEG responses across exposure groups (Figure 3). An in-depth overview of specific enriched pathways per exposure group is provided in Appendix A, with a list of all significantly enriched canonical pathways available in Appendix A.

With respect to ZnO, dissolved ZnCl_2_ enriched the most pathways at 2 h, while ZnO MPs showed the strongest pathway response at 24 and 48 h. At 2 h, ZnCl_2_ presented the largest number of enriched pathways, with 9 and 101 pathways at 5.1 and 8.5 µg/mL, while ZnO NPs and MPs presented 98 and 10 significantly enriched pathways at 5 µg/mL, respectively. At 24 h, ZnO MPs had the greatest number of enriched pathways (228 unique perturbed pathways at 5 µg/mL across low and high-fold-change exposure groups), with ZnCl_2_ presenting 25 and 177 enriched pathways at 5.1 and 8.5 µg/mL, respectively, and ZnO NPs enriching 149 pathways at 5 µg/mL. At 48 h, the number of enriched pathways was reduced compared to 24 h for all forms of Zn (Figure 3). Specifically, ZnO MPs exposure resulted in 199 total unique pathways at 5 µg/mL, which includes low and high-fold-change DEG groups; ZnCl_2_ exposed samples presented 36 and 146 enriched pathways at 5.1 and 8.5 µg/mL; and ZnO NPs presented the lowest number of enriched pathways with 95 pathways at 5 µg/mL.

With respect to NiO, dissolved Ni in the form of NiCl_2_ showed the most pronounced response at the pathway level compared to NiO NPs and MPs (Figure 3). At 2 h, no pathway enrichment was seen for any Ni compound. At 24 h, all Ni forms had concentration-dependent increases in enriched pathways, with NiCl_2_ presenting the largest number (58 and 93 pathways at 40 and 80 µg/mL, respectively), followed by NiO NPs (4, 46, and 80 significant pathways at 10, 25, and 50 µg/mL), and MPs (9 and 50 enriched pathways at 25 and 50 µg/mL). At 48 h, NiCl_2_ exhibited the most enriched pathways with 81 and 153 at 40 and 80 µg/mL, respectively, followed by NiO NPs with 10, 90, and 147 pathways for 10, 25, and 50 µg/mL, and NiO MPs with 17 and 37 pathways at 25 and 50 µg/mL concentrations. It is important to note that 1.6 µg/mL NiCl_2_, which corresponded to the extracellular dissolution of 50 µg/mL NiO NPs in DMEM at 48 h, showed no response at any timepoint tested.

For Al_2_O_3_, the largest number of enriched pathways was seen for AlCl_3_ when the concentration was expressed in terms of µg/mL Al (Figure 3). At 2 h, no significantly enriched pathways were seen for any Al form. At 24 h, both the Al_2_O_3_ NPs and MPs perturbed pathways at the highest concentration of 50 µg/mL, with 3 and 71 enriched pathways, respectively, for NPs and MPs, while AlCl_3_ enriched 1 and 6 pathways at 47 and 118 µg/mL. At 48 h, AlCl_3_ enriched 45 and 60 pathways at 47 and 118 µg/mL, followed by Al_2_O_3_ NPs with 36 and 46 enriched pathways at 25 and 50 µg/mL, and Al_2_O_3_ MPs showing the lowest response with 14 enriched pathways at 50 µg/mL.

Finally, for TiO_2_ particles, NPs presented a greater number of enriched pathways at each timepoint compared to MPs (Figure 3). At the earliest timepoint of 2 h, TiO_2_ NPs enriched 1, 2, and 5 pathways, and at 24 h, 33, 66, and 99 enriched pathways were seen for NPs at 25, 50, and 100 µg/mL concentrations, respectively. There were 6 and 27 enriched pathways seen for 50 and 100 µg/mL MP concentrations. At 48 h, a greater response was noted for the NPs, with 101, 162, and 262 significantly enriched pathways at 25, 50, and 100 µg/mL compared to 14 and 40 pathways for 50 and 100 µg/mL MPs.

### 2.5. Hierarchical Clustering of Pathway Responses

Hierarchical clustering was applied across all exposure groups and significantly enriched IPA canonical pathways in order to infer similarities and differences in response across the compounds tested (Appendix A). For this purpose, significantly enriched pathways were clustered based on their z-score, which is a metric used to denote directionality in pathway response (see methods). The previously published gene expression data from cells exposed to Cu forms [33] were included in this analysis. Samples that cluster close together are expected to enrich similar pathways with expression of DEGs showing similar directionality (upregulation or downregulation).

From the dendrogram in Figure 4, four main clusters are apparent and are separated mainly by the type of metal. Overall, the pathway responses seen across exposure conditions were largely MO form-specific, with metal salts and NPs inducing qualitatively similar responses, albeit with differences in timing and magnitude of response. With respect to ZnO materials, hierarchical clustering showed that ZnO NPs and ZnCl_2_ had similar transcriptional responses at 2 h (appear on the same node in cluster 3), while ZnO MPs were dissimilar. By 24 and 48 h, all Zn materials (ZnO NPs, MPs, and ZnCl_2_) enriched similar pathways with the same directionality and appeared in cluster 4. For CuO materials, clustering showed that CuO NPs had different pathway responses at 24 h and 48 h (present in cluster 1 (24 h) and cluster 4 (48 h)). Response to 7 µg/mL CuCl_2_ at 48 h was more akin to CuO NPs at 24 h than NPs at 48 h (Figure 4). Exposure to CuO MPs did not yield any enriched pathways with z-scores at any timepoint or concentration. Furthermore, there were no enriched pathways for CuO NPs or CuCl_2_ at 2 h. All Ni materials (NiO NPs, MPs, and NiCl_2_) showed similar pathway responses across time, with all exposure groups appearing within cluster 2 and no pathway enrichment seen at 2 h (Figure 4). With respect to Al materials, no pathway response was seen at 2 h for any compound, while at 24 h, only AlCl_3_ and Al_2_O_3_ MPs presented pathways with z-score directionality. At 24 h, the response was varied, with 118 µg/mL AlCl_3_ appearing in cluster 2 alongside NiO MP samples, whereas 50 µg/mL Al_2_O_3_ MPs appeared in cluster 3 alongside ZnO NP and ZnCl_2_ at 2 h. No pathways were enriched for Al_2_O_3_ NPs at 24 h. By 48 h, significantly enriched pathways and their directionality were similar between Al_2_O_3_ NPs and AlCl_3_ (all appeared in cluster 3), whereas 50 µg/mL Al_2_O_3_ MPs appeared outside of identified clusters. Finally, for TiO_2_ NPs and MPs, hierarchical clustering indicated that pathway responses between NPs and MPs were dissimilar. At 2 h, only 100 µg/mL TiO_2_ NPs samples presented enriched pathways and were grouped away from identified clusters. At 24 h, 25–100 µg/mL of TiO_2_ NPs appeared in cluster 2, whereas at 48 h, both 25 and 50 µg/mL of TiO_2_ NPs appeared in cluster 3, with 100 µg/mL of TiO_2_ NPs still present in cluster 2 on the same node as 24 h TiO_2_ NP samples. On the other hand, 100 µg/mL of TiO_2_ MPs at 24 h appeared in cluster 1, while at 48 h 50 µg/mL of TiO_2_ MPs appeared outside of identified clusters, and 100 µg/mL of MPs appeared in cluster 3.

### 2.6. HIF-1α Signaling as a Commonly Enriched Pathway across Metal Groups

All significantly enriched IPA canonical pathways were ranked by the number of samples showing enrichment to identify pathways commonly enriched across the exposure groups. Rank ordering of the top 10 most commonly enriched pathways (Table 2) showed the ‘HIF-1α Signaling’ canonical pathway as the most ubiquitously enriched across all exposures, with 35/58 samples showing pathway enrichment (a list of all samples showing enrichment of this pathway is found in Figure 5a, with a high resolution heat map available in Appendix A, and a pathway diagram found in Figure A1 and Figure A2 in Appendix B).

The ‘HIF-1α Signaling’ IPA pathway was the most commonly enriched across all MONPs (17/26 samples showing enrichment, Appendix A), while for MOMPs and metal chlorides it was the 2nd most commonly enriched pathway (8/17 and 10/15 samples showing enrichment, Appendix A). Clustering of DEGs associated with the ‘HIF-1α Signaling’ pathway showed treatment groups cluster mainly based on parent metal (Figure 5). Spearman’s correlation analysis showed that pathway coverage (number of DEGs / total genes in the pathway) of the ‘HIF-1α Signaling’ pathway was significantly negatively correlated with the percent viable cell density (*p* < 0.001; Spearman’s correlation coefficient = −0.905).

To identify DEGs from the ‘HIF-1α Signaling’ pathway that may be indicative of MONP toxicity, a weighted rank-ordering of all DEGs was conducted using scores calculated as follows: score = (average significant fold-change across samples) + (number of samples where the DEG fold-change > 5)*100. This weighting puts emphasis on DEGs, which show a high fold change in many samples as opposed to a very high fold change in a few samples.

The top ten ranked DEGs using the approach are listed in Figure 6. All Cu and Zn samples strongly upregulated the expression of heat shock protein A member 1A (*Hspa1a*), heat shock protein A member 1B (*Hspa1b*), and heme oxygenase 1 (*Hmox1*) (1.6–337 fold, −1.6–151 fold, and 4.3–48 fold, respectively). Conversely, *Hspa1a* and *Hspa1b* were not differentially expressed in Ni samples (with three minor exceptions), whereas solute family carrier member 1 (*Slc2a1*) and egl-9 family hypoxia-inducible factor 1 (*Egln1*) were both ubiquitously expressed with high fold-change (1.7–16 fold, 1.9–8.4 fold). For Al and Ti samples, all of the top 10 ranked DEGs showed fold-changes <5 fold in either direction. It should be noted that while Zn, Cu, and Ni materials upregulate the expression of adrenomedullin (*Adm*), both Al and Ti materials lead to downregulation of this gene. For matrix metallopeptidase 10 (*Mmp10*) and serpin family E member 1 (*Serpine1*), upregulation was seen for Zn, Cu, Ni, Al, and Ti materials, with the highest fold-changes noted in CuO NP-exposed samples (9.9–69 fold and 3.2–12 fold, respectively). The last gene was ras related dexamethasone induced 1 (*Rasd1*), which showed high fold-change expression in ZnO MP and ZnCl_2_-exposed samples at 24 and 48 h exclusively (7.6–49 fold).

### 2.7. Benchmark Concentration Modeling

Benchmark concentration modeling of viable cell density responses (Figure 7) and transcriptomic responses (Figure 8) was employed for comparative potency ranking of the MONPs, MOMPs, and metal chlorides used in this study, with the concentrations normalized to the amount of constituent metal in the exposure medium. Viable cell density was used instead of percent viability, as it more accurately captured the toxicity induced by Zn compounds compared to white and blue cell ratios in this study. Transcriptomic and cell density data from Cu exposures in Boyadzhiev et al., 2021 [33] were included in the analysis.

BMC modeling of viable cell densities was only attempted for 24 and 48 h timepoints due to the minimal response at the 2 h timepoint for all compounds. At 24 h, the materials were ranked as follows: ZnO NP ~ ZnO MP ~ ZnCl_2_ > CuO NP ~ NiCl_2_ > TiO_2_ NP ~ NiO MP ~ NiO NP ~ CuCl_2_ ~ Al_2_O_3_ MP ~ AlCl_3_ ~ TiO_2_ MP. At 48 h, a similar trend was observed, although the BMC ranges largely overlapped (Figure 7b). There were differences noted within the three forms of each metal compound at 48 h (Figure 7c). For Zn compounds, all three forms presented the same potency at 48 h, which was also seen for Al_2_O_3_ NPs and AlCl_3_, as well as for CuO NPs and CuCl_2_. In contrast, NiO NPs were more potent compared to NiCl_2_, while TiO_2_ NPs and NiO NPs were more potent than MP analogues.

With respect to the transcriptomic data, the transcriptional point of departure (tPOD) chosen was the 25th ranked gene for 2, 24, and 48 h timepoints (Figure 8). At 2 h, TiO_2_ NPs and NiO NPs were less potent than ZnO NPs, ZnO MPs, ZnCl_2_, CuO NPs, and CuO MPs at inducing transcriptional signaling. At 24 h, Zn compounds presented equivalent potency between the NP, MP, and ZnCl_2_ forms, an equivalency that was also displayed by the three forms of Ni and Al and the two forms of TiO_2_ (Figure 8d). In contrast, CuO NPs were more potent than MPs at inducing transcriptional signaling. By 48 h, Zn, Al, and Cu compounds presented the same potency trends as the 24 h timepoint, while NiCl_2_ was more potent than NiO NPs and NiO MPs at inducing transcriptional signaling, and TiO_2_ NPs presented greater potency than TiO_2_ MPs.

For both viable cell density and transcriptomic BMC modeling, the concentrations were also expressed in terms of specific surface area (SSA; cm^2^/mL), using experimentally measured SSA values for NPs and manufacturer-provided SSA values for MPs (Appendix A). The SSA-based BMC ranges at 24 h indicated MPs present equal or greater potency than NPs for reduction of viable cell density (ZnO MP > ZnO NP; NiO NP ~ NiO MP; TiO_2_ MP > TiO_2_ NP), with a similar trend for the induction of transcriptional response (ZnO NP ~ ZnO MP; CuO NP > CuO MP; NiO MP > NiO NP; Al_2_O_3_ MP ~ Al_2_O_3_ NP; TiO_2_ MP > TiO_2_ NP). By 48 h of exposure, substances showed compound-specific potency differences for reduction of viable cell density (ZnO MP > ZnO NP; NiO NP > NiO MP; TiO_2_ NP ~ TiO_2_ MP), while MPs generally presented a greater potency than NPs to induce transcriptional response (ZnO MP > ZnO NP; CuO NP > CuO MP; NiO MP > NiO NP; Al_2_O_3_ MP ~ Al_2_O_3_ NP; TiO_2_ MP > TiO_2_ NP).

## 3. Discussion

The solubility of MONPs in biological environments can have an impact on their potential to induce toxicity. Recent research has shown that size and concentration both affect the dissolution of MONPs, with MOMPs dissolving to a lesser extent [31,32]. In the present study, in vitro global transcriptomic analysis was applied to characterize the impact of solubility on cellular toxicity induced by MONPs over short exposure durations, using MOMPs and metal chloride salts as bulk and dissolved metal analogues. Overall, the results show that (1) extracellular dissolution was an important contributor to the toxicity of ZnO and CuO MONPs with high–moderate solubility in cell culture medium, while extracellular dissolution did not contribute to toxicity induced by NiO and Al_2_O_3_ MONPs with low–negligible solubility; (2) responses were generally specific to the parent metal (Zn, Cu, Ni), with MONPs and metal salts inducing similar transcriptional responses albeit with differences in magnitude and timing, while MOMPs showed less consistency; (3) signaling through the ‘HIF-1α Signaling’ pathway was a common response underlying the toxicity of MONPs, as well as their bulk and dissolved metal analogues; however, the DEGs within the pathway for each metal differed, implying differences in underlying mechanisms and the eventual toxicity; and (4) BMC-derived relative potency ranking of apical and transcriptomic endpoints indicated similar toxicity for ZnO NPs, MPs, and Zn ions, whereas differences were observed in the case of MONPs that have lower solubility in biological media.

### 3.1. The Impact of Solubility on the Toxicity of MONPs

The contribution of extracellular dissolution to the response induced by MONPs depended on the solubility of the material. For ZnO NPs, which display high solubility and instantaneous dissolution in cell culture medium at low concentrations (Appendix A), an equivalent concentration of ZnCl_2_ induced a similar cytotoxic and transcriptomic response as early as 2 h post-exposure, which markedly increased for ZnCl_2_ at 24 and 48 h post-exposure (Figure 1, Figure 2 and Figure 3). Other in vitro studies in the past have reported similar levels of acute cytotoxicity for dissolved Zn and ZnO NPs when the concentration was normalized to the constituent metal [36] or markedly higher toxicity in response to dissolved Zn exposure [37]. For the moderately soluble CuO NPs (Appendix A), extracellular dissolution contributed to cellular toxicity, as 7 µg/mL CuCl_2_ induced transcriptional signaling at 48 h, which clustered alongside 24 h CuO NP samples [33] (Appendix A, Figure 4). However, the results showed a lag in time at which the cytotoxicity or significant transcriptional perturbance became apparent post-exposure to CuCl_2_. Also, extracellular dissolution alone was insufficient to account for the magnitude of the responses seen for NPs at 24 and 48 h (described in detail in Boyadzhiev et al., 2021 [33]), implying a role for the particulate form of CuO NPs in the observed toxicity. In the case of NiO and Al_2_O_3_, NPs that showed low-negligible solubility in cell culture medium, concentrations of metal chlorides representing possible extracellular dissolution of NPs (1.6 and 2.4 µg/mL, respectively) did not result in loss of viability or induce transcriptional response at any timepoint assessed (Figure 1, Figure 2 and Figure 3). For NiO NPs, this indicates that toxicity is induced either through cell-material interactions or through particle internalization and intracellular dissolution, rather than from exposure to ions via extracellular dissolution. For Al_2_O_3_ NPs, however, the result is less straightforward. Upon addition of dissolved AlCl_3_ to FE1 cell culture medium, an insoluble precipitate was quickly formed at a concentration of 1 mg/mL (Appendix A). Similar analyte loss during control experiments was reported by Avramescu et al., 2023 [32] at a lower AlCl_3_ concentration of 1 mg/L. Considering the observed analyte loss, the authors suggested that the solubility reported may be considered an ‘apparent solubility’, which represents the portion of released Al that is not matrix bound [32]. Furthermore, particle formation has been noted by others when conducting in vitro exposures using AlCl_3_ as well as within simulated intestinal fluid [38,39]. The formation of aluminum phosphate and calcium phosphate precipitate has been reported when AlCl_3_ was introduced into A549 cell culture medium [40]. The results of these studies, as well as the results reported here, imply that loss of dissolved Al within neutral or slightly basic biological environments is an anticipated behavior and may be caused by precipitation, surface adsorption, and/or interactions with proteins. Moreover, the present study indicates that extracellular dissolution based on ‘apparent solubility’ does not contribute to responses induced by Al_2_O_3_ NPs. In summary, while extracellular dissolution was a factor contributing to the observed toxicity of certain MONPs with high-moderate solubility, it was not a significant factor impacting the toxicity of MONPs with low-negligible solubility.

Based on transcriptional profiling, MONPs and dissolved salts of the same parent metal (e.g., Ni) perturbed similar pathways at the same timepoints of the assessment when equivalent metal concentrations, or concentrations equivalent to high levels of NP dissolution (≥ 50 %) were used (Figure 4 and Appendix A). With respect to bulk MP analogues, responses were less consistent, with ZnO and NiO MPs clustering alongside their NP and dissolved analogues, but with Al_2_O_3_ and TiO_2_ MPs showing less consistency. Detailed analysis of significantly enriched pathways (Appendix A) indicated a conserved stress and cell death response for ZnO NPs, MPs, and dissolved Zn beginning at the earliest post-exposure timepoint of 2 h and increasing in severity up to 48 h at equivalent Zn concentrations (4 µg/mL Zn). However, it is important to note that the number of enriched pathways at the highest concentration decreased for all forms of Zn from 24 to 48 h. For NiO NPs, MPs, and dissolved Ni, pathways implicated in hypoxia signaling beginning at 24 h were seen, with concentration- and time-dependent increases in pathways involved in cell stress, death, and immune response (Appendix A). It was commonly seen that 50 µg/mL NPs (40 µg/mL Ni) clustered close to 80 µg/mL NiCl_2_ (20 µg/mL Ni) at the same timepoints (Figure 4), implying a similar pathway response between the two forms but with an increased magnitude and breadth of transcriptional induction for dissolved Ni. For Al_2_O_3_ NPs, the pathway response was negligible for all concentrations of NPs and AlCl_3_ at 24 h (<10 significant pathways), with 48 h responses for both materials featuring mainly aberrant wound healing responses, immune signaling, and disease pathways noted at the highest exposure concentrations of 50 µg/mL of NPs (26.5 µg/mL Al) and 118 µg/mL of AlCl_3_ (13.25 µg/mL Al). Recent reports examining MONPs alongside dissolved metal equivalents in human cell lines support this view, with ZnO NPs and ZnCl_2_ showing a transcriptionally similar response in A549 cells after 24 h of exposure [41]; and NiO NPs and NiCl_2_ exposures both showing cytotoxicity and genotoxicity in BEAS-2B cells following both chronic (6-week) and acute (24 h) in vitro exposures, with NiCl_2_ inducing the most DEGs after 6 weeks of exposure compared to NiO NPs at the same concentration of Ni [42,43]. While some commonalities were observed, for example, enrichment of ‘HIF-1α Signaling’ across all samples, the results showed that mechanistically, each metal type differs from the other, and thus, eventual toxicity may also be different.

### 3.2. ‘HIF-1α Signaling’ as a Common Underlying Response to MONP Toxicity

Based on shared enriched canonical pathways, it could be seen that ‘HIF-1α Signaling’ was the most commonly enriched pathway across all exposures except for Al_2_O_3_ MPs (Table 2), with DEG responses clustering based on parent metal (Figure 5b). Higher coverage in this pathway (more DEGs) was strongly associated with decreases in the density of viable cells in this study (Figure 5c). Signaling through the ‘HIF-1α Signaling’ pathway showed MONPs with low-negligible solubility in cell culture medium of low acute toxicity (Al_2_O_3_, TiO_2_) altered the expression of lower numbers of DEGs in general, with lower fold-change differences in comparison to toxic MONPs with higher solubility (ZnO, CuO) (Appendix A).

The ‘HIF-1α Signaling’ pathway is a stress-induced signaling pathway that controls the expression of over 100 downstream genes [44]. Cellular processes directly controlled by HIF-1α include angiogenesis, metabolism, iron transport and cycling, extracellular matrix remodeling, and cell survival and differentiation [44,45]. Cellular response to hypoxia is controlled by both oxygen and iron availability, and dysregulation in iron homeostasis is known to induce a hypoxic response in a similar manner to low oxygen [46]. Responses associated with hypoxia include a shift in energy generation away from oxygen-intensive pathways (e.g., mitochondrial cellular respiration) and towards anoxic processes (e.g., glycolysis), as well as the induction of stress responses such as autophagy, endoplasmic reticulum stress, oxidative stress, and unfolded protein responses [47,48,49,50]. High levels of oxidative stress can lead to oxidative damage of biomolecules such as lipids, proteins, and nucleic acids, which, in turn, leads to DNA damage and unfolded protein response induction, which are two main mechanisms of MONP toxicity [13]. Furthermore, the accumulation of unfolded proteins in cells can result in endoplasmic reticulum stress, while disturbances in the endo-lysosomal network can cause autophagic stress, both of which have been associated with the toxicity of MONPs such as ZnO, CuO, and iron oxide nanoparticles [19]. The ability to monitor these important stress pathways, along with HIF-1α’s direct association with cellular homeostasis and specifically metal homeostasis, makes this pathway an attractive candidate as an in vitro biomarker of MONP toxicity. While all MONPs responded by inducing ‘HIF-1α Signaling’, the specific DEGs and consequently, associated processes varied. Oxidative stress and unfolded protein responses were commonly seen in Zn, Cu, and Ni-exposed cells and, to a lesser extent, in TiO_2_ NP exposed cells; however, important pathways implicated in glycolysis were for the most part not enriched (Appendix A). NiO NPs exhibit low-negligible solubility in cell culture medium [32] and moderate-low solubility in acidic environments [51], with dissolved Ni shown to induce a state of cellular hypoxia akin to the hypoxia-mimetic compound cobalt chloride [52,53]. Transcriptional induction of hypoxia was seen for NiO NPs, MPs, and NiCl_2_, which all enriched and activated ‘HIF-1α Signaling’, in addition to activation of glycolysis for all Ni compounds and inactivation of oxidative phosphorylation for NiO NPs and NiCl_2_ (Appendix A, Appendix A). It should be noted that while all metal compounds lead to the enrichment of ‘HIF-1α Signaling’, only NiO NPs, MPs, and NiCl_2_ lead to activation of this pathway based on z-score directionality (Figure 5a).

The ‘HIF-1α Signaling’ IPA canonical pathway consists of 198 genes, based on IPA content version 81348237. Using weighted rank-ordered genes from samples where the ‘HIF-1α Signaling’ pathway was enriched, the top 10 DEGs were selected with high fold-change induction across the sample set (Figure 6). These genes consist of *Hmox1*, *Hspa1a*, *Hspa1b*, *Mmp10*, *Adm*, *Serpine1*, *Slc2a1*, *Egln1*, *Rasd1*, and *Hk2*. The genes *Hmox1, Hspa1a,* and *Hspa1b* are implicated in iron recycling, response to oxidative stress, iron homeostasis, and protein folding and are transcriptionally responsive to reactive oxygen species, heavy metal exposure, as well as homeostatic disruption [54,55,56]. Both HSP genes were highly expressed in response to Zn and Cu exposures, but not Ni, Al, or Ti, whereas *Hmox1* was highly expressed in Zn, Cu, and Ni samples. Two genes implicated in glucose uptake and metabolism, *Slc2a1* and *Hk2*, showed the highest fold-change in 25 and 50 µg/mL at 48 h of NiO NPs and 40 and 80 µg/mL at 24 and 48 h of NiCl_2_ exposed groups. Next was *Egln1*, which showed high fold-change induction in Ni-exposed samples. This gene encodes a prolyl hydroxylase domain-containing protein used to post-translationally control the stability of HIF1A [45]. The gene *Adm* encodes the multi-functional peptide adrenomedullin, which has recently been shown to have increased expression in mouse lung epithelium in hypoxic conditions [57]. This gene was upregulated in Zn, Ni, and Cu-exposed samples and downregulated in Al and Ti-exposed samples. The next two genes, *Mmp10* and *Serpine1,* encode a matrix metalloprotease and a serine protease inhibitor, respectively, and show upregulation in response to Zn, Cu, Ni, Al, and Ti exposures. Differential expression of *Serpine1* has been associated with pulmonary disease [58,59], and increased expression has been noted in the lungs of mice exposed to carbon nanotubes and TiO_2_ [21,60]. Similarly, *Mmp10* has been implicated in pulmonary inflammation and the development of pathologies such as emphysema and chronic obstructive pulmonary disease [61], with high levels of MMP10 expression noted in the lungs of rats under hypoxic stress [62]. The final DEG in the list was *Rasd1*, which only showed expression in ZnO MP and ZnCl_2_-exposed samples at 24 and 48 h, but with a high fold change. The gene *Rasd1* encodes a RAS small GTPase protein involved in many mammalian signaling cascades, including iron cycling and calcium signaling [63]. These 10 genes involved in HIF-1α signaling showed strong induction in response to highly toxic MONPs, such as ZnO, CuO, and NiO NPs, but minimal perturbation in low-responding Al_2_O_3_ and TiO_2_ NPs. Therefore, it is posited that their expression levels may serve as putative in vitro markers of exposure to toxic MONPs.

### 3.3. Relative Potency Ranking of Apical and Transcriptomic Points of Departure

In order to delineate differences in potency amongst MONPs, MOMPs, and metal chlorides with respect to their ability to induce cytotoxicity and transcriptomic signaling, BMC modeling was employed (Figure 7, Figure 8, Appendix A). Compound-specific differences in potency were observed. When concentrations were expressed in terms of the mass volume of the constituent metal, there were no observable differences in potency amongst the three Zn compounds at 24 or 48 h with respect to transcriptomic response and cytotoxicity. CuO NPs with moderate solubility showed a greater ability to induce cytotoxicity than dissolved Cu (as CuCl_2_) at 24 h, but this difference in potency disappeared at 48 h. Transcriptionally, NPs were more potent than MPs at both 24 and 48 h, with no similar comparison possible for CuCl_2_ due to the paucity of concentrations used. NiO NPs with low-negligible solubility presented mixed potency compared to MPs and dissolved Ni, with NiCl_2_ presenting greater potency than NPs and MPs at inducing cytotoxicity at 24 h, and equivalent potency to induce transcriptional response. However, by 48 h, NiO NPs presented greater potency to induce cytotoxicity compared to both NiCl_2_ and NiO MPs, whereas NiCl_2_ was more potent at inducing transcriptional responses at this timepoint. For Al_2_O_3_ NPs with low-negligible solubility, no difference in potency was seen for cytotoxicity at 48 h between NPs and AlCl_3_, a trend similarly seen for NPs, MPs, and AlCl_3_ transcriptomic exposures at 24 and 48 h. It must be reiterated, however, that dissolved Al precipitated in cell culture medium (Appendix A) and therefore does not represent Al fully in solution. Overall, MOMPs tended to present equivalent or lower cytotoxic and transcriptional potencies in comparison to both MONPs and dissolved metals. Together, the results of this BMC modeling provide support for the notion that cellular toxicity resulting from exposure to ZnO NPs with high-moderate solubility in cell culture medium is comparable to exposure to fully dissolved ions. However, MONPs of lower solubility show compound-specific trends in potency between nano, bulk, and dissolved forms.

The BMC concentration was also expressed in terms of the SSA of each particle, which takes into account the available surface atoms for interaction, and which is proposed as a more appropriate dose metric for MONPs with negligible solubility compared to mass dose [64]. For the NPs used in this study, experimentally derived SSA values were available from recent publications [65,66]. However, for MPs, only the manufacturer-provided ranges were available (See Section 4 for details). Using the mean SSA estimates for MPs, it was observed that MOMPs were either more potent or showed equivalent potency to MONPs for both endpoints, with the exception of CuO NPs, which were more potent than CuO MPs at inducing transcriptional signaling at 24 and 48 h, and NiO NPs, which were more potent than MPs at inducing reductions in viable cells at 48 h (Appendix A). The use of manufacturer-provided SSA values for MOMPs precluded a more precise potency comparison between NPs and MPs, but the current results concur with reports indicating that differences in potency between NPs and MPs can at least partially be explained by the larger SSA of NPs [67].

### 3.4. Implications for Risk Assessment Strategies

There is a pressing need to derive consistent information concerning the impacts of physicochemical properties on the toxicity mechanisms of nanomaterials, such as MONPs, in order to allow for a timely and accurate assessment of the risks these materials pose to both human and environmental health. Solubility has been used as a case-study method for binning materials for environmental risk assessment by determining which species (particulate, dissolved, or both) should be the focus of the risk assessment [35]. Furthermore, solubility has been proposed as an aspect of a tiered testing approach to the human health risk assessment of nanomaterials in Canada [68]. The present study provides evidence of compound-specific effects of solubility on MONP toxicity, with a key conclusion that there are limits on the applicability of solubility in read-across and grouping strategies for the purposes of risk assessment. That is, solubility cannot be considered a sole determinant of the toxicity of MONPs, with further systematic studies required to identify other properties that act in parallel or in conjunction to influence toxicity.

Through transcriptomic analysis, this study identified signaling through the ‘HIF-1α Signaling pathway’ as a conserved response across not only MONPs, but also metal chlorides and most MOMPs, with the number of differentially expressed genes in this pathway closely tied to cytotoxicity. Due to the association of HIF-1α signaling with cell death, cellular homeostasis, as well as with key stress processes such as oxidative stress, protein damage, and material turnover (autophagy), key genes in this pathway can be explored for use as prognostic in vitro biomarkers for acutely toxic MONPs in a tiered testing approach. Fold-change induction in these reporters can prioritize MONPs for toxicity testing using more advanced in vitro techniques or toxicity verification by animal testing methods.

## 4. Materials and Methods

### 4.1. Nanoparticles, Microparticles, and Metal Chlorides

A total of 4 MONPs, 4 MOMPs, and 3 metal chloride salts were used to probe the impact of solubility over time on the toxicity of MONPs, with the Cu materials assessed in Boyadzhiev et al., 2021 [33] used for comparison (Table 3). All nanoparticles were purchased commercially and selected to have a pristine surface without coatings or functionalizations and similar manufacturer-reported primary particle sizes (<50 nm). The microparticles were also commercially procured and selected to have a reported mean primary particle size range of around 1 µm. In certain cases where particles of approximately 1 µm were not available, larger particles with a reported size of 5 µm were purchased.

### 4.2. Cell Culture

The immortalized mouse alveolar epithelial cell line, FE1, was utilized for toxicity testing in this study. These cells retain characteristics of both type I and type II alveolar epithelial cells with a doubling time of 18.7 ± 2 h [69,70] and used for genotoxicity testing and global response characterization of both nanomaterials and chemicals [34,35,71].

FE1 cells were maintained in phenol-red-containing DMEM/F12 medium (DMEM/F12 (1:1); Cat#: 11320-033, Life Technologies, Burlington, ON, Canada), with 2% fetal bovine serum (FBS; Cat#: 12483-020, Life Technologies, Burlington, ON, Canada), 1 ng/mL of human epidermal growth factor (Cat#: PHG0311, Life Technologies, Burlington, ON, Canada), 100 U/mL of penicillin G, and 100 µg/mL of streptomycin (Cat#: 15140-122, Life Technologies, Burlington, ON, Canada), in a climatized incubator (37 °C, 5% CO_2_). For exposures, phenol-red-free medium was used (Cat#: 21041-025, Life Technologies, Burlington, ON, Canada).

### 4.3. Particle Characterization

To assess the particle size of the Al_2_O_3_ and NiO NPs and the morphology of the Al_2_O_3_ and NiO MPs, TEM and SEM imaging were respectively employed. For Al_2_O_3_ and NiO NPs, a JEM-2100F Field Emission TEM (JEOL Ltd., Peabody, MA, USA) was used to collect 10 non-overlapping micrographs of the dry particles. From each image, the length and width of 9–20 well-defined particles were measured for a total of 100–200 particles per MONP. The size distributions of each of the particles were plotted as histograms in SigmaPlot 12.5 (Systat Software Inc., San Jose, CA, USA), and the mean length and width were reported with a standard deviation, along with the aspect ratio. With respect to Al_2_O_3_ and NiO MPs, a JSM-7500F Field Emission SEM (JEOL Ltd., Peabody, MA, USA) was used to collect 10 non-overlapping micrographs of the dry particles. For ZnO and TiO_2_ materials investigated here, TEM and SEM analyses were conducted and described in Boyadzhiev et al., 2022 [34].

Particle size and zeta potential of each MONP suspension were carried out using DLS and ELS as described in Boyadzhiev et al., 2021 [33]. MONP stock suspensions in UltraPureTM DNase/RNase-Free distilled water (dH_2_O) (Cat#: 10977015, Life Technologies, Burlington, ON, Canada; 5 mg/mL, details in Section 4.4) were diluted to 50 µg/mL in phenol red-free DMEM/F12 (1:1) medium + 2% FBS (described in Section 4.2), and aliquots were analyzed by DLS/ELS using a Zetasizer Nano ZSP (Malvern Panalytical, Westborough, MA, USA). Mean hydrodynamic diameter (Dh, nm; as a measure of aggregate size) and PDI, as well as zeta potential (ZP, mV: a measure of the surface charge of the particles), were determined for each aliquot. Each measurement was conducted in triplicate for the calculation of the mean and standard deviation.

### 4.4. Preparation of Stock Particle Suspensions and Metal Chloride Solutions

Stock dispersions of MONPs and MOMPs were prepared at 5 mg/mL in dH_2_O. The suspensions were sonicated using a Branson Digital Ultrasonics Sonifier™ SFX550 (Branson Ultrasonics, Brookfield, CT, USA) equipped with a half inch disruptor horn probe with extension and removable flat tip according to procedures described in Avramescu et al., 2019 and Avramescu et al., 2023 [32,72]. The sonicated particle stock dispersions in water were diluted in phenol-red-free cell culture medium to 0.5–100 µg/mL concentrations and subsequently used for exposures. For the metal chloride exposure series, all compounds were first dissolved in dH_2_O at stock concentrations of 1–5 mg/mL and subsequently diluted in phenol-red-free cell culture medium for cell exposures.

### 4.5. Concentration Selection

The exposure concentrations of MONPs, MOMPs, and metal chlorides used in this study are provided in Table 4. Multiple metrics are included to allow for intercomparison between the materials. The average SSA from the range provided by the manufacturer was used to calculate the SSA-based concentration of MOMPs. With respect to ZnO NPs, concentrations in the 0–25 µg/mL range were initially selected based on 48 h toxicity reported in other lung epithelial cell lines such as A549 and BEAS2B, which show high levels of cytotoxicity above this range [73,74]. The NiO and Al_2_O_3_ NP exposure concentrations ranged from 0–50 µg/mL, while for TiO_2_ NPs, concentrations in the 0–100 µg/mL range were selected due to the lower cytotoxicity generally reported using this material. For each metal oxide, equivalent concentrations of MOMPs were used.

With respect to concentrations used for each of the metal chloride exposures, selection was based on the 48 h dissolution of the MONP in acellular conditions within cell culture medium [31,32], with the aim of recapitulating the level of dissolved metal ions as an estimate of extracellular particle dissolution. In the context of ZnO NPs, particles display 95% dissolution at 10 µg/mL, and the concentrations of ZnCl_2_ used correspond to 100% particle dissolution. With respect to both NiO and Al_2_O_3_ NPs, 48 h dissolution was ~1% in cell culture medium, and the lowest concentrations used for NiCl_2_ (1.6 µg/mL) and AlCl_3_ (2.37 µg/mL) correspond to a 1% dissolution of 50 µg/mL of NPs. With respect to Al_2_O_3_ NPs, it is worth noting that analyte loss was reported in AlCl_3_ control experiments, and so the 1% solubility represents ‘apparent solubility’ based on the fraction of Al that is not matrix bound [32]. The upper concentrations of 40 and 80 µg/mL for NiCl_2_ and 47 and 118 µg/mL for AlCl_3_ represent higher levels of solubility with equivalency to particle concentrations (Table 4). Due to the insoluble nature of TiO_2_ NPs, no dissolved metal equivalent was used for this particle type.

### 4.6. Cell Exposures, Phase Contrast Imaging, and Sample Collection

Cellular exposures to MONP, MOMP, and metal chlorides were conducted in 3–4 biological replicates under submerged culture conditions. The day preceding exposures, cells were seeded at a density of 130,000 cells / well in 6-well plates and incubated overnight (37 °C, 5% CO_2_). The next day, cells of ~20% confluence were exposed to 3–4 concentrations of ZnO, NiO, Al_2_O_3_, and TiO_2_ MONPs, MOMPs, and metal chlorides in cell culture medium (Table 4). For all experiments, an unexposed blank medium served as a negative control. After 2, 24, and 48 h of exposure, the medium from each sample was removed, cells were washed with 0.5 mL of PBS, and phase contrast images were acquired using an EVOS XL microscope (Thermofisher Scientific, Waltham, MA, USA). The cells were then dissociated with 0.25% trypsin-EDTA (Cat#: 25200-056, Life Technologies, Burlington, ON, Canada) and resuspended in fresh phenol-red-free cell culture medium. An aliquot from each suspension was used for trypan blue exclusion staining. The remaining suspensions were pelleted (8000 rpm, 10 min, 4 °C), resuspended in 0.5 mL PBS, pelleted again (8000 rpm, 10 min, 4 °C), and then frozen at −80 °C for downstream analysis.

### 4.7. Trypan Blue Exclusion Staining

Trypan blue exclusion staining was performed as described in Boyadzhiev et al., 2021 [33], and Boyadzhiev et al., 2022 [34], with some modifications. In brief, for each sample, 10 µL of cell suspension was combined with 10 µL of trypan blue dye (Cat#: 15250-061, Life Technologies, Burlington, ON, Canada) and incubated at room temperature for 5–10 min. The mixtures were then loaded onto a hemocytometer for manual white (viable) and blue (non-viable) cell counting. The percent viability was calculated as the ratio of white and blue cells in a sample. All data were arcsine normalized via the equation arcsine (%viability/100). Statistically significant differences compared to time-matched medium controls were determined through a two-way ANOVA with a Dunnett’s post-hoc in SigmaPlot 12.5 (Systat Software Inc., San Jose, CA, USA).

A second metric was calculated based on the density of viable cells for MONP, MOMP, and metal chloride-exposed samples compared against the density of viable cells for time-matched medium controls (percent viable cell density; (density_of_viable_cells_SAMPLE/density_of_viable_cells_MEDIUM)*100)). This metric takes into account differences in cell density at harvest between exposed and control cells. For this calculation, the viability data from Boyadzhiev et al., 2021 [33] were included for comparison with the other examined metal oxides. The square root normalized densities of viable cells for MONP, MOMP, and metal chloride-exposed samples were compared in SigmaPlot 12.5 (Systat Software Inc., San Jose, CA, USA) against the square root normalized density of viable cells for time-matched medium controls using a two-way ANOVA with a Dunnett’s post-hoc in the case of a significant difference.

### 4.8. RNA Extraction, Purification, and Integrity Analysis

Frozen FE1 cell pellets harvested from 2–48 h MONP, MOMP, and metal chloride exposures to 2–48 h were lysed using TRIzol (Cat#: 15596026, Invitrogen, Burlington, ON, Canada), and then RNA was extracted and purified using the Direct-zol RNA miniprep kits (Cat#: R2052, Zymo Research, Irvine, CA, USA) according to the manufacturer’s protocol. The RNA concentration was measured using a NanoDrop One spectrophotometer (Thermofisher Scientific, Waltham, MA, USA), while RNA integrity was assessed using both an Agilent 2100 BioAnalyzer (Agilent Technologies, Inc., Santa Clara, CA, USA) and an Agilent 4200 Tapestation System (Agilent Technologies, Inc., Santa Clara, CA, USA) according to the manufacturer’s instructions. All extracted samples that showed RNA integrity numbers > 6.5 were used for gene expression analysis.

### 4.9. Microarray Hybridization

Microarray hybridization was conducted as described in Boyadzhiev et al., 2021 [34]. A randomized block design was used for each sample group [75]. For each sample, 200 ng of purified RNA as well as 200 ng of universal mouse reference RNA were reverse transcribed into cDNA, which was subsequently transcribed into cyanine 5-CTP-labeled (sample) and cyanine 3-CTP-labeled (reference) cRNA. Next, 300 ng of cyanine 5-CTP-labeled sample RNA and 300 ng of cyanine 3-CTP-labeled reference were mixed and hybridized onto 8 × 60K Agilent SurePrint G3 Mouse Gene Expression v2 Microarray slides (Agilent Technologies, Inc., Santa Clara, CA, USA) in a hybridization oven for 17 h (65 °C, 10 rpm). The next day, slides were washed according to the manufacturer’s instructions and scanned on an Agilent G4900DA Microarray Scanner and an Agilent G2505B Microarray Scanner (Agilent Technologies, Inc., Santa Clara, CA, USA). Data from the resulting images were extracted using the Agilent Feature Extraction 11.01.1 software (Agilent Technologies, Inc., Santa Clara, CA, USA).

### 4.10. Statistical Analysis of Microarray Data

Normalization and statistical analysis of microarray data were conducted as in Boyadzhiev et al., 2021 [33]. First, the data were normalized using the LOWESS method (locally weighted scatterplot smoothing), after which statistically significant differential gene expression was determined using a microarray analysis of variance (MANOVA) in ‘R’ [76]. To test for treatment effects, the Fs statistic was used, while *p*-values were estimated using the permutation method with adjustments made for multiple comparisons using the false discovery rate multiple testing correction [77]. For differentially expressed genes (DEGs), the fold-change was based on the least-square means [78]. Genes were considered differentially expressed if they exhibited adjusted *p*-values < 0.05 and |fold-change| ≥ 1.5 (*n* = 2–4). The microarray datasets used in this manuscript can be found in the NCBI gene expression omnibus, under the accession number GSE246159.

### 4.11. Ingenuity Pathway Analysis

For all samples, canonical pathway analysis was conducted on the probe list (Differential expression cut-offs: *p*  ≤  0.05, |fold-change| ≥ 1.5) using Ingenuity Pathway Analysis (IPA) (content version 81348237, license from Qiagen). While Cu samples were analyzed using IPA in Boyadzhiev et al., 2021 [33] (microarray dataset available from the NCBI GEO database under accession number GSE161017), the samples were reanalyzed due to content differences between the subsequent IPA database updates in order to allow for comparative hierarchical clustering with the four metal oxides examined in this study. In addition, with respect to the ZnO MP samples, there were too many significant molecules (>8000) to conduct an enrichment using the standard cut-offs. Thus, the ZnO MP probe lists were subdivided into 2 sets: the low fold-change set and the high fold-change set. The low list contained transcripts with fold-changes between 1.5 and 2-fold in either direction, while the high list contained all transcripts with a fold-change greater than or equal to 2 in either direction. The IPA enrichment was conducted on both lists separately. Pathways were considered to be significantly enriched if the −log(*p*-value) > 1.3 and the total amount of DEGs was ≥3. Similarly, enriched pathways were considered to be activated if the Z-score ≥ 2, and inactivated if the Z-score ≤ −2. 

In order to identify clusters of exposure groups with similar transcriptional responses, hierarchical clustering was conducted using the z-score of significantly enriched IPA canonical pathways as the clustering metric. The z-score is a statistical measure used to define how closely the expression of DEGs in a pathway corresponds to the literature-derived gene expression patterns indicative of pathway (in)activation [79]. A value ≥ 2 indicated strong concordance with expected expression patterns for activation of the pathway, while values ≤ −2 indicated strong negative concordance or inactivation of the pathway. Values between −2 and 2 indicated mixed signaling or no readily apparent state of activation or inactivation. For all samples where a pathway was non-significant, the z-score was adjusted to 0. This was to ensure a complete data matrix for clustering. Average linkage clustering was conducted using Cluster 3.0 [80], using uncentered Pearson’s correlation as the similarity measure. The resulting heatmap was visualized using Treeview 1.2.0 [81].

### 4.12. Benchmark Concentration Modeling

BMC modeling was conducted using both viability and microarray data. Viable cell densities (cells/cm^2^) were utilized for BMC modeling of viability data using BMDS Online2 [82]. The benchmark response factor used was hybrid extra risk, at a benchmark response of 0.5 (50% increase over control), using normal and nonconstant variance settings. In each case, the ‘Viable–Recommended’ model was used [83], unless unavailable, in which case the ‘Questionable’ model with the lowest AIC without model saturation and a definite BMC interval was used. BMCs were reported alongside the lower 95th percentile BMCL and the upper 95th percentile BMCU values.

Log2 normalized fluorescence ratios from microarray experiments were input into BMDExpress2 for benchmark concentration modeling. Similar cutoffs were used as in Rowan-Caroll et al., 2021 [84], including pre-filtering with the Williams’ Trend test with 100 permutations, a 1.5-fold-change cutoff, and an unadjusted *p*-value < 0.05. Probes that passed this initial filter were modeled using the linear, hill, power, polynomial (2°, 3°), and exponential (2nd, 3rd, 4th, 5th) functions to determine their BMCs (benchmark response = 1 standard deviation relative to controls, confidence level = 0.95, power function restricted to ≥1, max 250 iterations, 300 s max model compute time). The resulting BMC data and probe information were exported from BMDExpress2 and further filtered in Excel. First, all probes with a best-fit *p*-value < 0.1 and/or BMCU/BMCL ≥ 40 were removed. Next, probes were removed if the best BMC > the highest tested concentration. Finally, all probes without a matching entrez ID and gene symbol were removed. The final probe list was collapsed down to gene symbol level, and the best BMC, BMCL, and BMCU values were summarized by their median for genes with multiple matching probes. For the purposes of relative potency ranking, the 25th-ranked gene was used as the tPOD.

## Figures and Tables

**Figure 1 ijms-25-00529-f001:**
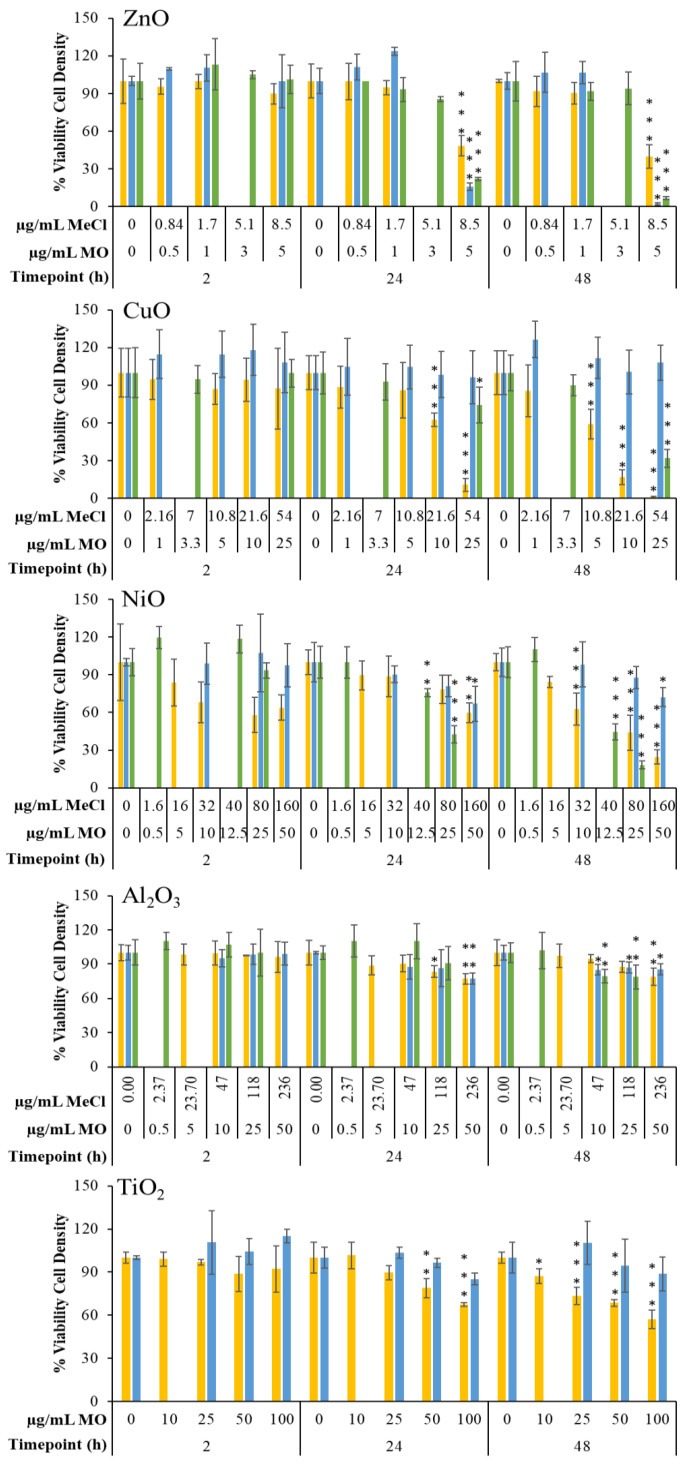
Percent viable cell density of FE1 cells following 2, 24, and 48 h of exposure to MONPs, MOMPs, and metal chlorides compared to time-matched medium controls. Error bars indicate the standard deviation (*n* = 3–4). Graphs were labeled based on the type of metal oxide. Yellow: MONPs. Blue: MOMPs. Green: metal chloride salts. MO: metal oxide. MeCl: metal chloride. Statistical significance against time-matched medium controls was determined through a 2-way ANOVA with a Dunnett’s post-hoc. *: *p* < 0.05. **; *p* < 0.01. ***; *p* < 0.001.

**Figure 2 ijms-25-00529-f002:**
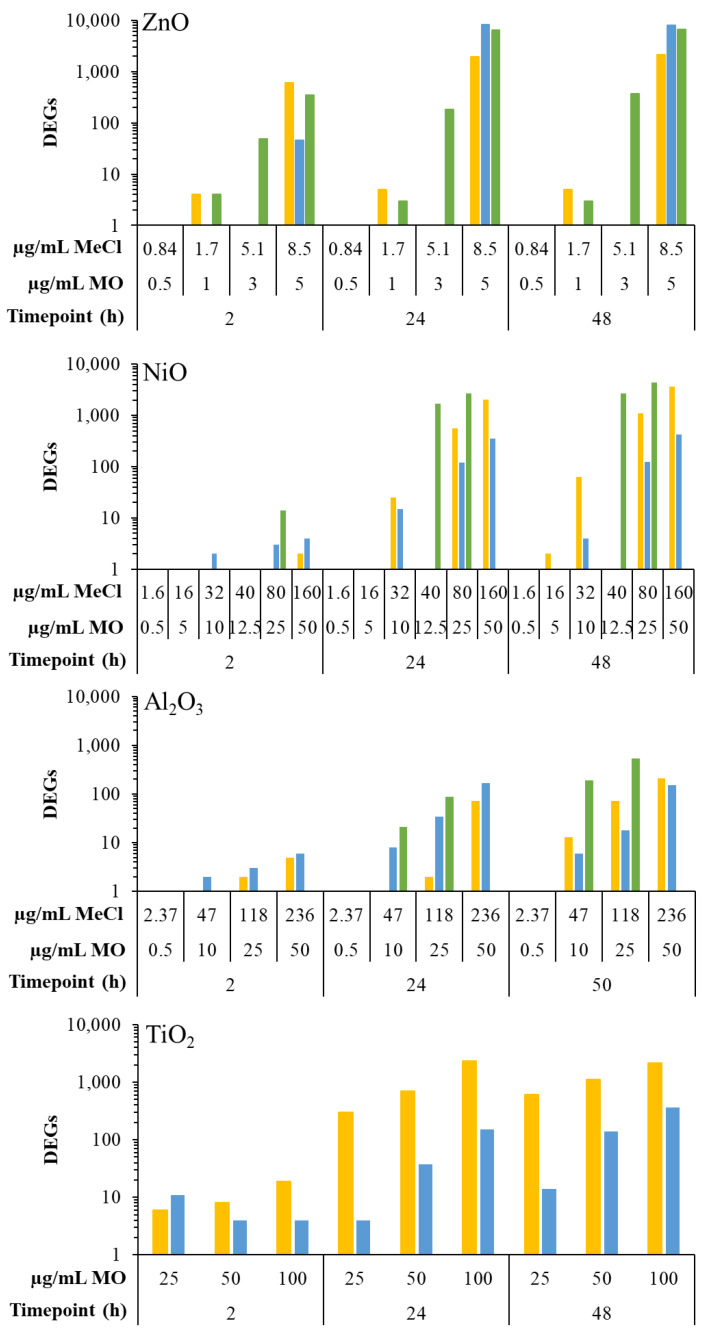
Total number of differentially expressed genes following 2–48 h exposure to MONPs, MOMPs, and metal chloride salts. Graphs were labeled based on the type of metal oxide. Yellow: MONPs. Blue: MOMPs. Green: metal chloride salts. MO: metal oxide. MeCl: metal chloride.

**Figure 3 ijms-25-00529-f003:**
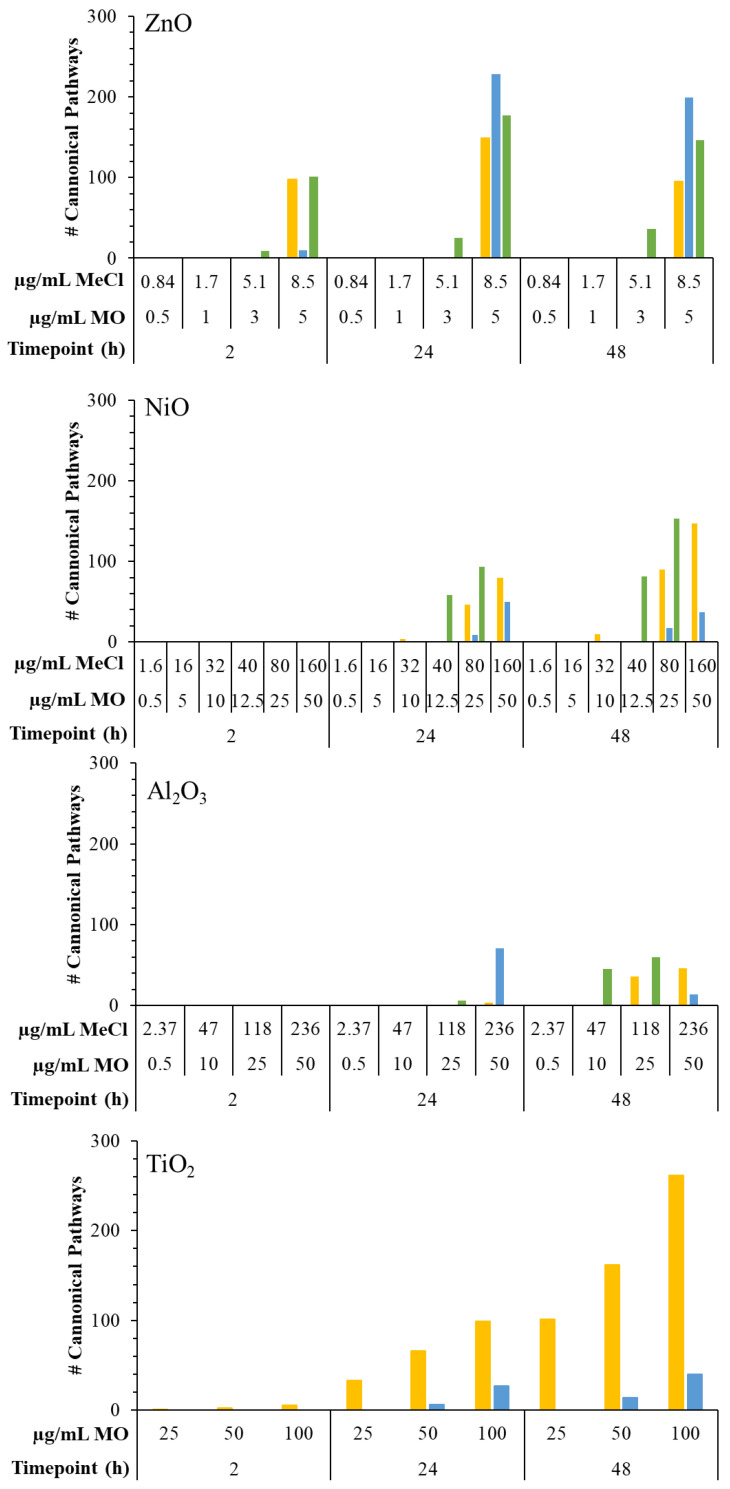
Total number of perturbed IPA canonical pathways following 2–48 h exposure to MONPs, MOMPs, and metal chloride salts. Graphs were labeled based on the type of metal oxide. Yellow: MONPs. Blue: MOMPs. Green: metal chloride salts. MO: metal oxide. MeCl: metal chloride. Significantly enriched pathways were combined in the case of ZnO MPs from enrichment of both the low fold-change and high fold-change datasets (see Section 4).

**Figure 4 ijms-25-00529-f004:**
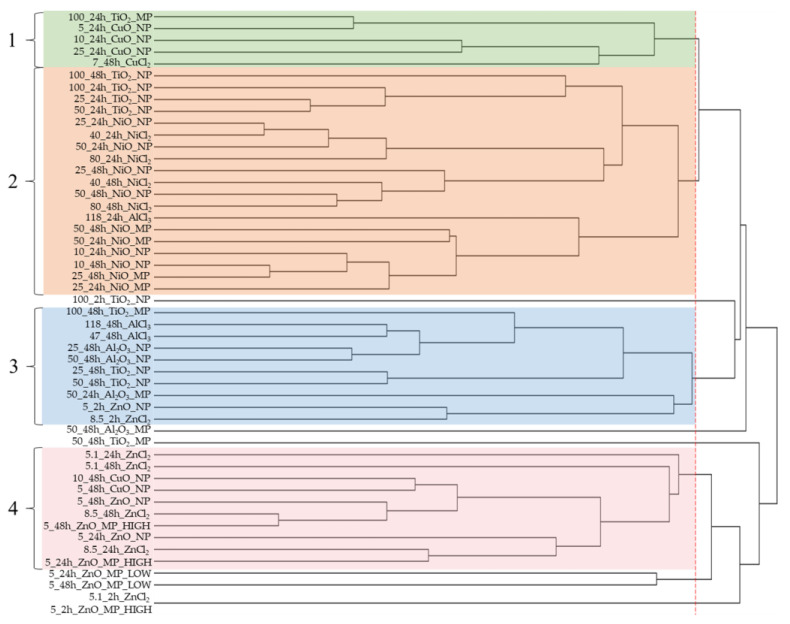
Hierarchical clustering of metal oxide and metal chloride-induced canonical pathway perturbation The clustering was conducted on the z-scores of significantly enriched pathways. The red dashed line indicates where the dendrogram was cut to produce groupings. Each grouping was numbered and highlighted with a colored box.

**Figure 5 ijms-25-00529-f005:**
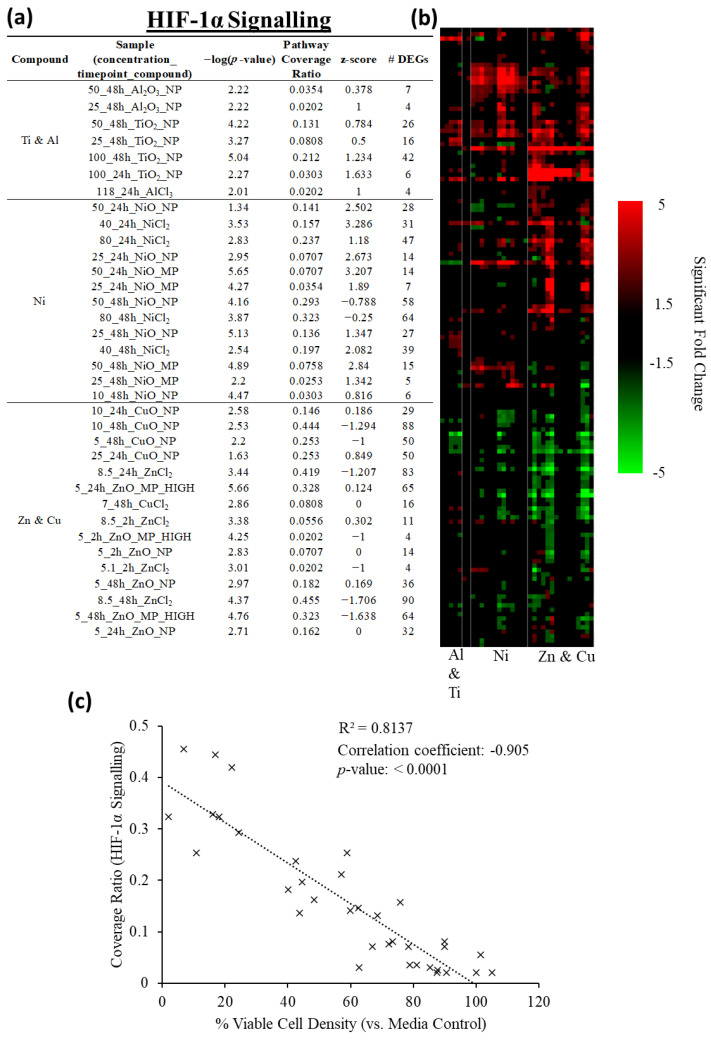
Perturbation of the ‘HIF-1α Signaling’ pathway by MONPs, MOMPs, and metal chloride samples from 2–48 h. (**a**) Number of DEGs, pathway coverage ratio (# DEGs/total genes in the pathway), *p*-value, and z-score for all samples for which this pathway was enriched. (**b**) Heatmap showing clustering of samples and DEGs associated with the pathway. Lines indicate groups formed from the sample dendrogram. (**c**) Percent viable cell density plotted against the coverage ratio of the ‘HIF-1α Signaling’ pathway for samples where the pathway was significantly enriched. A Spearman’s correlation was conducted, with the resulting correlation coefficient and *p*-value displayed and a trendline fit to the graph.

**Figure 6 ijms-25-00529-f006:**
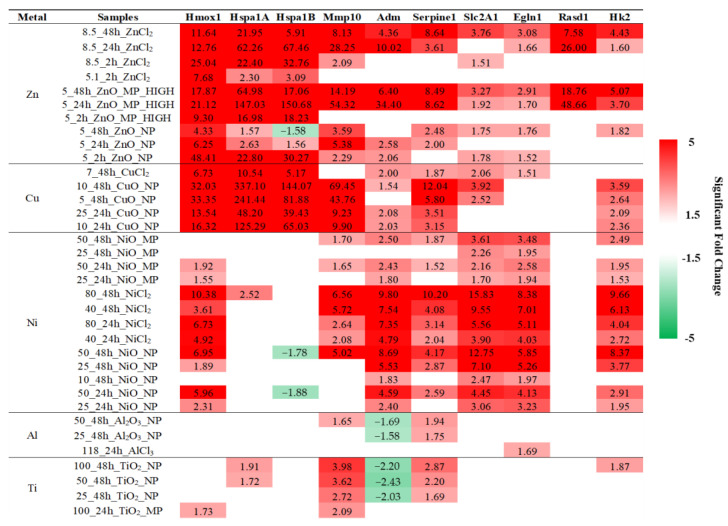
Heatmap showing the differential expression of the top-10-ranked DEGs from the ‘HIF-1α Signaling’ pathway across all samples for which this pathway is significantly enriched. The values in each cell indicate the fold change over the control. The left-most DEG is the top-ranked, whereas the right-most DEG is the bottom-ranked. Blank cells indicate no significant differential expression. Red: increased fold change. Green: decreased fold change.

**Figure 7 ijms-25-00529-f007:**
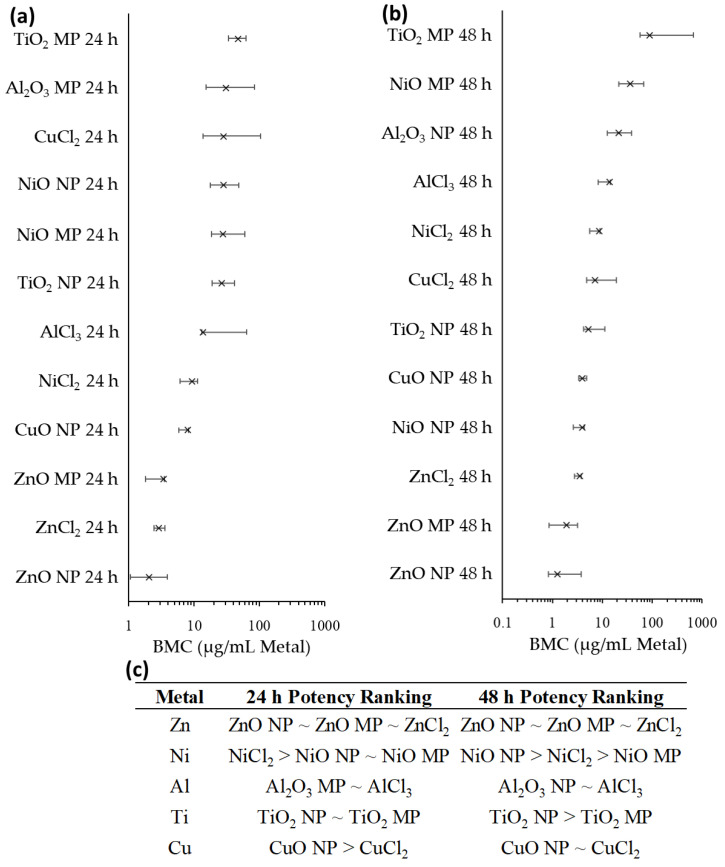
BMDS-based BMC modeling of viable cell density reduction following (**a**) 24 h and (**b**) 48 h exposure to metal oxides and metal chlorides, with (**c**) differences in potency for each metal variety. Benchmark response: 0.5 (hybrid risk). The ‘x’ indicates the BMC. Left and right bars indicate lower and upper 95% confidence intervals around the BMC, respectively. Potencies for different compounds are considered equivalent if the confidence intervals overlap.

**Figure 8 ijms-25-00529-f008:**
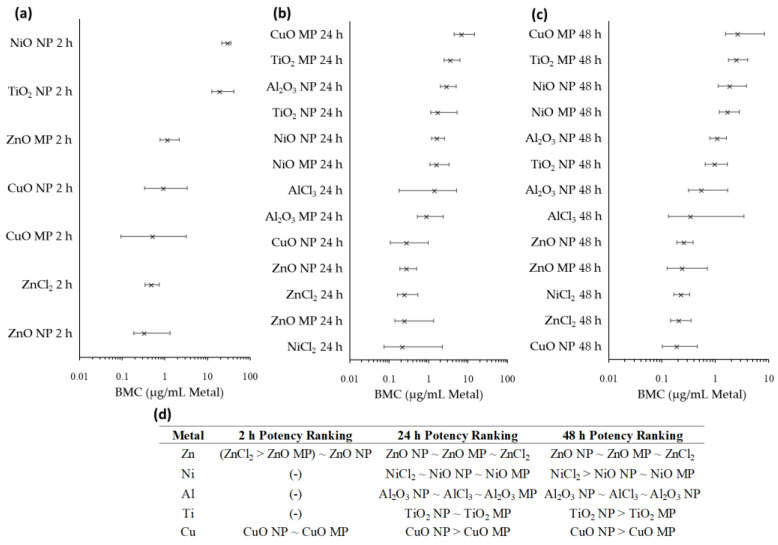
The transcriptomic-based BMC determined through BMDExpress2 BMC modeling for (**a**) 2 h, (**b**) 24 h, and (**c**) 48 h metal oxide and metal chloride exposures, with (**d**) differences in potency for each metal variety. Benchmark response: 1 (standard deviation). The ‘x’ indicates the BMC. Left and right bars indicate lower and upper 95% confidence intervals around the BMC, respectively. Potencies for different compounds are considered equivalent if the confidence intervals overlap.

**Table 1 ijms-25-00529-t001:** MONP characterization in DMEM + 2% FBS. All particles were suspended at 50 µg/mL for DLS and ELS characterization. Values in parentheses indicate ± standard deviation (*n* = 3–4). PPS: primary particle size [length × width]. PDI: polydispersity index.

MONP	PPS Measured (nm)	Aspect Ratio	PDI	Aggregate Size (Dh, nm)	Zeta Potential (mV)
ZnO	23.9 × 19.4 (7.2 × 5.5) ^a^	1.23 (0.17) ^a^	0.52 (0.05)	346 (46)	−9.5 (0.1)
NiO	27.3 × 21.9 (10.3 × 7.91)	1.25 (0.20)	0.33 (0.01)	178 (7)	−12.3 (0.6)
Al_2_O_3_	23.9 × 10.7 (11.8 × 6.86)	2.63 (1.40)	0.44 (0.01)	276 (16)	−12.2 (0.2)
TiO_2_	26.8 × 20.8 (8.9 × 6.8) ^a^	1.30 (0.26) ^a^	0.23 (0.06)	421 (31)	−11.5 (0.2)
CuO	64.8 × 45.9 (47 × 28) ^b^	1.39 (0.39) ^a^	0.174 (0.009) ^b^	182 (1.4) ^b^	−10.8 (0.5) ^b^

^a^ Data from Boyadzhiev et al., 2022 [34]. ^b^ Data from Boyadzhiev et al., 2021 [33].

**Table 2 ijms-25-00529-t002:** The top ten most commonly enriched IPA canonical pathways following 2–48 h of exposure to MONPs, MOMPs, and metal chlorides.

Pathway	Total Number of Samples Where Pathway Is Enriched
HIF1α Signaling	35/58
Pulmonary Fibrosis Idiopathic Signaling Pathway	31/58
Role of Macrophages, Fibroblasts and Endothelial Cells in Rheumatoid Arthritis	31/58
ID1 Signaling Pathway	30/58
Aryl Hydrocarbon Receptor Signaling	29/58
Hepatic Fibrosis / Hepatic Stellate Cell Activation	29/58
IL-10 Signaling	28/58
Hepatic Fibrosis Signaling Pathway	28/58
LPS/IL-1 Mediated Inhibition of RXR Function	28/58
Osteoarthritis Pathway	27/58
Tumor Microenvironment Pathway	27/58

**Table 3 ijms-25-00529-t003:** Nanoparticles, microparticles, and metal chlorides used for toxicity investigation in this study. NP: nanoparticle. MP: microparticle. PPS: primary particle size. SSA: specific surface area. N/A: not available.

Metal	Material	Catalogue Number (Manufacturer)	PPS ^a^ [nm]	SSA (Average) [m^2^/g]
Zn	ZnO NP	US3580 (US Research Nanomaterials Inc., Houston, TX, USA)	35–45	35 ^b^
	ZnO MP	US1003M (US Research Nanomaterials Inc., Houston, TX, USA)	1000	2–15.8 (8.9) ^a^
	ZnCl_2_	Z0152-100G (Sigma Aldrich, Oakville, ON, Canada)		
Ni	NiO NP	US3355 (US Research Nanomaterials Inc., Houston, TX, USA)	15–35	38.7 ^c^
	NiO MP	US1014M (US Research Nanomaterials Inc., Houston, TX, USA)	5000	5–20 (12.5) ^a^
	NiCl_2_ • 6H_2_O	N6136-100G (Sigma Aldrich, Oakville, ON, Canada)		
Al	Al_2_O_3_ NP	544833 (Sigma Aldrich, Oakville, ON, Canada)	<50	129 ^b^
	Al_2_O_3_ MP	1331DL (Sky Spring Nanomaterials Inc., Houston, TX, USA)	400–1500	110
	AlCl_3_ • 6H_2_O	A0718-500G (Sigma Aldrich, Oakville, ON, Canada)		
Ti	TiO_2_ NP	NIST 1898 (National Institute of Standards and Technology, Gaithersburg, MD, USA)	19 nm (Anatase) 37 nm (Rutile)	55.5 ^a^
	TiO_2_ MP	US1017M (US Research Nanomaterials Inc., Houston, TX, USA)	1500	5–8 (6.5) ^a^
Cu ^d^	CuO NP	544868 (Sigma Aldrich, Oakville, ON, Canada)	<50	4.6 ^b^
	CuO MP	US1140M (US Research Nanomaterials Inc., Houston, TX, USA)	5000	4–6 (5) ^a^
	CuCl_2_ • 2H_2_O	C3279-100G (Sigma Aldrich, Oakville, ON, Canada)		

^a^ Manufacturer provided. ^b^ Data from Bushell et al., 2020 [65]. ^c^ Data from Kunc 2022 [66]. ^d^ Material evaluated in Boyadzhiev et al., 2021 [33].

**Table 4 ijms-25-00529-t004:** Summary of FE1 exposure concentrations for each of the 4 different types of metal oxides and 3 metal chlorides assessed, in addition to Cu materials previously examined. MO: metal oxide. MONP: metal oxide nanoparticle. MOMP: metal oxide microparticle. MeCl: metal chloride. Me: Metal. (-): particle of metal chloride concentration not assessed. N/A: not applicable/available.

Metal	Concentration [µg/mL MO]	Concentration [cm^2^/mL MONP]	Concentration [cm^2^/mL MOMP]	Concentration [µg/mL MeCl]	Concentration [µg/mL Me]	Concentration [µM Me]
Zn	5.00	1.75	0.45	8.50	4.00	61.44
	(-)	(-)	(-)	5.10	2.45	37.42
	1.00	0.35	0.09	1.70	0.80	12.29
	0.50	0.18	0.04	(-)	0.40	6.14
Ni	50.00	19.35	6.25	(-)	40.00	669.41
	25.00	9.68	3.13	80.00	20.00	334.70
	(-)	(-)	(-)	40.00	10.00	167.40
	10.00	3.87	1.25	(-)	8.00	133.88
	5.00	1.94	0.63	(-)	4.00	66.94
	(-)	(-)	(-)	1.60 ^a^	0.40	6.69
Al	50.00	64.50	N/A	(-)	26.50	980.78
	25.00	32.25	N/A	118.00	13.25	490.39
	10.00	12.90	N/A	47.00	5.30	196.16
	5.00	6.45	N/A	(-)	2.65	98.08
	(-)	(-)	(-)	2.37 ^b^	0.27	9.81
Ti	100.00	55.50	6.50	N/A	60.00	1252.10
	50.00	27.75	3.25	N/A	30.00	626.05
	25.00	13.88	1.63	N/A	15.00	313.02
	10.00	5.55	0.65	N/A	6.00	125.20
Cu	25.00	1.15	1.25	54.00	20.00	314.29
	10.00	0.46	0.50	(-)	8.00	125.72
	5.00	0.23	0.25	(-)	4.00	62.86
	(-)	(-)	(-)	7.00 ^c^	2.60	41.06
	1.00	0.05	0.05	(-)	0.80	12.57

^a^ Represents the expected amount of dissolved Ni based on the 1% dissolution of 50 µg/mL NiO NPs at 48 h [31]. ^b^ Represents the expected amount of dissolved Al based on the 1% dissolution of 50 µg/mL Al_2_O_3_ NPs at 48 h [32]. ^c^ Represents 2.64 times the expected amount of dissolved Cu from 10 µg/mL CuO NPs at 48 h [33].

## Data Availability

The data presented in this study are available on request from the corresponding author.

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
