# Peer review of "Toxicity of Metal Oxide Nanoparticles: Looking through the Lens of Toxicogenomics"

_ijms, 2023, doi:10.3390/ijms25010529_

Round 1

Reviewer 1 Report

Comments and Suggestions for Authors

Very interesting manuscript looking at various metal oxide nanoparticles and their toxicity and transcriptomic response.

Figure 1 was a slight struggle to interpretate – Perhaps the authors want to make this easier to read.

It isn’t clear what method of exposure was used – I’m assuming it was a submerged exposure, but the author should make this slightly more obvious. They should also comment on why this exposure method was used and why the cells weren’t used at an air-liquid interface.

Author Response

Please see attached response document.

Reviewer 2 Report

Comments and Suggestions for Authors

This study investigates the impact of solubility on the toxicity of metal oxide nanoparticles (MONPs) using FE1 mouse lung epithelial cells exposed to zinc oxide, nickel oxide, aluminum oxide, and titanium dioxide. The findings reveal that material solubility alone does not govern MONP toxicity; instead, the conserved transcriptional response through the 'HIF-1α Signalling' pathway emerges as a key biomarker, allowing for prioritization of MONPs for further evaluation and guiding targeted testing strategies. The study is presented well and data is shown as per claim made in the study, however, there are major knowledge gap, which needs to be addressed, as show in my comments below. Addressing these points will contribute to the overall clarity, depth, and impact of the paper on wound microbiota and its influence on wound healing-

  1. Introduction: Clarification of Research Gap- It would be beneficial to explicitly state the existing knowledge gap that this study aims to address. Clearly articulate what is not known or not well understood in the current literature regarding the impact of solubility on the toxicity of metal oxide nanoparticles (MONPs) citing a recent report https://doi.org/10.1016/j.scitotenv.2022.160503 with the opening sentence in introduction section.
  2. Introduction: Definition of Toxicogenomics- Provide a concise definition or explanation of toxicogenomics early in the introduction to ensure that readers, including those not familiar with the term, can readily understand its relevance to the study.
  3. Experimental Design: Rationale for Cell Line Choice- Justify the choice of FE1 mouse lung epithelial cells as the experimental model. Explain why this cell line is particularly suitable for studying the toxicity of MONPs and how its characteristics align with the objectives of the study.
  4. Methods: Controls and Replicates- Explicitly state the positive and negative controls used in the experiments to ensure the validity of the results. Furthermore, elaborate on the number of replicates performed for each condition to establish the robustness of the findings. Add catalogue number of all reagents/materials/assay kits and model number of instruments used for characterization with software details. Also, add cell acquisition number or ATCC of all primary cells/cell lines used in the study. This information is critical to replicate the experiments.
  5. Methods: Transcriptomic Analysis- Elaborate on the specific methods employed for the microarray analysis. Provide information on data normalization procedures, statistical methods used for identifying differentially expressed genes (DEGs), and any adjustments made for multiple comparisons.
  6. Results: Viability Assessment- Include additional information on the kinetics of viability changes over the exposure period. A time-dependent analysis of viability could offer insights into the dynamics of cellular response to MONPs.
  7. Experimental Design: Solubility Variation- Provide more details on how solubility variations were achieved for each MONP. Specify the concentrations used and the rationale behind selecting these concentrations. Additionally, discuss any challenges encountered in controlling and verifying the solubility of the nanoparticles citing a latest study https://doi.org/10.21203/rs.3.rs-3358190/v1 along with the sentence ´ of which is particle solubility in biological environments´ on page 3 along line 103.
  8. Results: Comparative Analysis with Copper Oxide- When comparing the results with previously published data on copper oxide exposure, discuss the similarities and differences observed. Highlight any insights gained from this comparative analysis and its implications for understanding MONP toxicity.
  9. Results: Material-Specific DEGs- Provide a detailed breakdown of the material-specific DEGs, emphasizing their potential implications for toxicity mechanisms. This could involve discussing individual genes and their known roles in cellular processes.
  10. Discussion: Linking HIF-1α Signaling to MONP Toxicity- Clearly articulate the rationale behind selecting the HIF-1α signaling pathway as a biomarker for MONP exposure and toxicity. Discuss the biological significance of this pathway in the context of oxidative stress and unfolded protein responses.
  11. Discussion: Implications for Risk Assessment- Extend the discussion to address the broader implications of the findings for risk assessment of MONPs. How can the identified biomarkers inform future risk assessment strategies, and what are the potential regulatory implications?

Comments on the Quality of English Language

Minor editing of English language required

Author Response

Please see attached response document.

Reviewer 3 Report

Comments and Suggestions for Authors

The manuscript "Toxicity of Metal Oxide Nanoparticles: Looking Through the Lens of Toxicogenomics" is well-written and effectively presented, making it suitable for publication. However, I have a few suggestions:

  1. 1. Include some key results in the abstract.
  2. 2. Specify the doubling hours of the cell line and the percentage confluence of cells during the experiment in the methods section.
  3. 3. Justify the choice of using trypan blue for cell viability assay despite its limitations and potential counting errors.
  4. 4. Enhance the last paragraphs of the discussion section by summarizing the bottom line of the work based on the obtained results.

Author Response

Please see attached response document.

Round 2

Reviewer 2 Report

Comments and Suggestions for Authors

accept